**Determining the hierarchical order by which the variables of sampling season, dust outbreaks occurrence, and sampling location, can shape the airborne bacterial communities in the Mediterranean basin**

Riccardo Rosselli[1], Maura Fiamma[2], Massimo Deligios[2], Gabriella Pintus[3], Grazia Pellizzaro[3], Annalisa Canu[3], Pierpaolo Duce[3], Andrea Squartini[4], Rosella Muresu[5], Pietro Cappuccinelli[2]

[1]Department of Biology, University of Padova, Via Ugo Bassi 58/b, 35131 Padova, Italy [2]Department of Biomedical Sciences-University of Sassari, Italy, [3]Institute of Biometeorology-National Research Council (IBIMET-CNR), Italy, [4]Department of Agronomy Animals, Food, Natural Resources and Environment, DAFNAE, University of Padova, Viale dell'Università 16, 35020 Legnaro (Padova) Italy, [5]Institute of Animal Production Systems in Mediterranean Environments-National Research Council (ISPAAM-CNR),Italy.

Correspondence: Andrea Squartini (squart@unipd.it)

Keywords: Airborne microbiota, dust outbreaks, Mediterranean, Sardinia

**Abstract**

An NGS-based taxonomic analysis was carried out on airborne bacteria sampled at ground level in two periods (May and September) and two opposite localities on the North-South axis of the Sardinia Island. Located in a central position of the Mediterranean basin, Sardinia constitutes a suitable outpost to reveal possible immigration of bacterial taxa during transcontinental particle discharge between Africa and Europe. With the aim of verifying relative effects of dust outbreaks, sampling period and sampling site, on the airborne bacterial

community composition, we compared air collected during dust-carrying meteorological events to that coming from wind regimes not associated to long-distance particle lifting. Results indicated that: (a) a higher microbial diversity (118 orders vs 65) and increased community evenness were observed in the campaign carried out in September in comparison to the one in May, irrespective of the place of collection and of the presence or absence of dust outbreaks. (b) During the period of standard wind regimes without transcontinental outbreaks a synchronous, concerted succession of bacterial communities across distant locations of the same island, accompanied as mentioned by a parallel rise in bacterial diversity and community evenness appears to have occurred. (c) changes in wind provenance could transiently change community composition in the locality placed on the coast facing the incoming wind, but not in the one located at the opposite side of the island; for this reason the community changes brought from dust outbreaks of African origin are observed only in the sampling station exposed to south; (d) the same winds, once proceeding over land appear to uplift bacteria belonging to a common core already present over the region, which dilute or replace those that were associated with the air coming from the sea or conveyed by the dust particulate, explaining the two prior points. (e) the hierarchy of the variables tested in determining bacterial assemblages composition results: sampling period >> ongoing meteorological events > sampling location within the island.

## 1 Introduction

With a total volume evaluated of $4.5x10^{18}$ $m^3$, terrestrial lower atmosphere represents the most extended potential biome, followed by water, $1.3x10^{18}$ $m^3$ (Gleick, 1993), and by soil with $6.2x10^{16}$ $m^3$ (estimated on the basis of the deeper subsurface living bacteria currently described Szewzyk *et al.*, 1993). Concerning atmosphere, microbial cells and propagules, embody a particularly suitable conformation to take advantage of air utilization as an environment for

survival and dispersion. Their movement can be favored by a natural mobile reservoir of physical solid carriers represented by the air-dispersed particulate matter. Such particles range between 0.2 and 10 um in size (Bernstein *et al.* 2004) and average loads of 1-100 ug $m^{-3}$ (Williams *et al.* 2002, Van Dingenen *et al.* 2004). It has been estimated that more than 5000 Tg of sea salt (Tegen *et al.* 1997) and 1000-2000 Tg of soil particles, passively uplifting and transporting live cells are released every year in the atmosphere giving rise to a widely heterogeneous material conveyed from different sources (Guang *et al.* 2009; Mc Tainsh 1989, Knippertz *et al.* 2009).

The tropical African and Asiatic belts (Prospero *et al*. 2002, Schepansky *et al.* 2007), represent two amongst the major airlift dust sources (http://www.who.int/). Several studies underline that this phenomenon strongly contributes to a cosmopolitan microbial distribution (Favet *et al.* 2013, Griffin 2008, Yang *et. al.* 2008, Wainwright *et al.* 2003, Smith *et al.* 2010). Moreover, the correlation between specific bacterial clades and particle size (Polimenakou *et al.* 2008) opened new hypotheses on differential dispersion of taxa in relation to the dust features. High amount of bacterial 'newcomers' have been pointed out in air samples collected in occasions of foreign dust outbreaks (Maki *et al.* 2014, Rosselli et al., 2015). Immigrant microorganisms classification (Sànchez de la Campa *et al.* 2013) and their effects on an autochthonous ecosystem have also been reported (Peter *et al.* 2014, Shine *et al.* 2000). Evidences of a correlation between aerosol-related biodiversity and seasons (Gandolfi *et al.* 2015) underlines the natural complexity related to this process, suggesting that effects may vary also depending on climatic periodicity. Marked seasonal patterns in airborne microbiota have also been reported in long term studies (Cáliz *et al.*, 2018). The genes that are specific to communities of bacteria inhabiting the atmosphere, referred to as aeolian lifestyle, have been studied by metagenomics approaches and include UV-induced DNA damage repair, cell aerosolization, aerotaxis, and thermal resistance (Aalismail et al., 2019).

Europe-Mediterranean air circulation routes offer an interesting case study when focusing on airborne bacteria. The system can be represented as a multidirectional network in which biological components and weather conditions are closely related (Lelived *et al.* 2002). Extending for more than 30 degrees of latitude above the subtropical belt, Europe is crossed by middle-latitude and equatorial atmospheric systems. Mathematical models suggest that a considerable part of the air mass movements has a Northern, Atlantic source in response to the pressure generated by the Azores high (Littmann, 2000). Southern winds from Africa, prone to carry desert sand, and potentially microbes, can be determined by specific climate conditions (Kostopoulou and Jones 2007, Benkhalifa et al, 2019). It has been estimated that, as a consequence, 80-120 Tg of dust per year are transported across the Mediterranean towards Europe (d'Almeida 1986; Dulac *et al.* 1996), reaching the higher troposphere layers (Alpert *et al.* 2004) and spilling over, until the far-Northern sides of the continent (Franzèn *et al.* 1991). In order to track the biodiversity of these airways, the Italian island of Sardinia was chosen as ideal observatory point to collect airborne bacteria moving inside and outside Europe. Located in the middle of the Mediterranean Sea, this landmass is separated from Italy, France, Spain and Africa coastal baselines by distances of 120, 150, 230, and 100 nautical miles (NM) respectively (Fig. 1). Its geographical position facilitates the displacement of western high- and low-pressure air masses coming from Gibraltar and becoming the first and the last frontier for microbes entering or leaving Europe, respectively. In a prior study (Rosselli et al. 2015), we described a core microbiome in the bacteria cast upon the Sardinia island under different wind regimes through analyses of DNA from deposited particles. The analysis compared the trans-Mediterranean airflow with that of winds from Europe, and pinpointed a number of taxa which have records in clinical infections. In that investigation the sampling dates were all concentrated in a single period of six days (in February) and some variations of the airborne biota were observed in response to the opposite wind

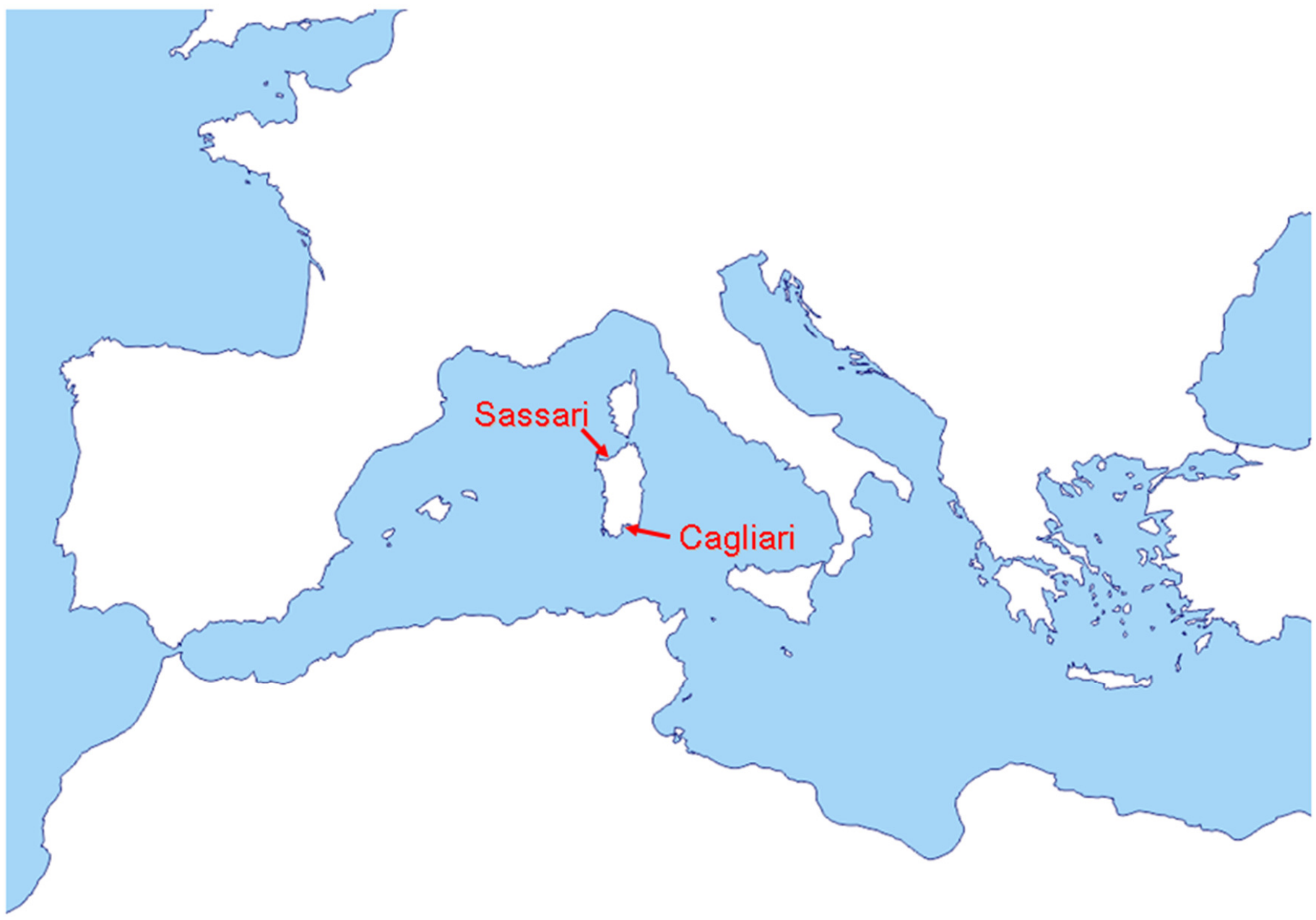

**Fig. 1** Mediterranean area with Sardinia Island detail and sampling locations Sassari and Cagliari.

directions. However, the most remarkable evidence was a prevailing constancy of the microbial composition in spite of the changing winds provenances. In the present study instead we analyzed a series of events featuring a starting dust outbreak, a 109 days-long period devoid of dust-carrying winds, and a second dust outbreak. The analyses were performed in two oppositely located stations: Cagliari, on the South-East side of Sardinia, facing the African side, and Sassari in the North-West, i.e. farthest from the dust-carrying winds. The sampled particulate was analyzed by NGS sequencing of the amplified 16S rRNA genes. The main goal of the project was to verify in which hyerarchical order the different variables of (a) sampling period, (b) occurrence of dust-carrying outbreaks, and (c) sampling location, could act in determining airborne bacterial communities composition.

## 2. Materials and Methods

### 2.1 Meteorological monitoring

Surveillance of the weather trends and conditions to anticipate dust outbreaks from Africa towards Sardinia and winds of interest was performed by routine checking of the MODIS satellite data and Meteosat imagery combined with the SKIRON forecasting model (Nickovic *et al.* 2001).

Europe daily synoptic conditions were analyzed on the weather charts available from the www.eurometeo.com and www.metoffice.gov.uk websites.

The origin and the trajectory of the dust carried by winds towards Italy were inferred by the NOAA HYSPLIT model (Hybrid Single Particle Lagrangian Integrated Trajectory Model) (Draxler *et al.* 2014; Rolph 2014).

Monitoring was aimed at predicting two distinct conditions: i) North-African high-pressure nuclei favoring Southern winds suitable to carry and deposit dust over Sardinia (dust-enriched events); and ii) North-European high-pressure nuclei, determining northern winds referred to as 'Controls' (dust-negative events).

In addition, PM10 concentration (particulate matter with a diameter of less than 10 µm) and meteorological data registered by the ARPAS (Regional Environmental Protection Agency of Sardinia) monitoring stations were taken into consideration in relation to the arrival of African air masses.

Information about wind direction and intensity (every 10 minutes), temperature and humidity (once per hour) were downloaded by the ISPRA website (http://www.mareografico.it/) and two sampling stations located in Cagliari (39.21°N, 9.11°E) and Sassari - Porto Torres (40.84°N,

8.40°E). Data covered a 7 months time-lapse, from March to September 2014, in order to obtain a nearly annual view to focus within the main weather instability period.

## 2.2 Sampling

Samples were collected on Teflon filters (Sartorius Stedim Biotech) by using a Skypost Tecora apparatus (compliant to the European legislation 96/62/gmeCE) processing 39 liters of air per minute. For each sample, date and atmospheric conditions are reported and fully described in the Results chapter and Supplementary Materials.

A one-day filtering step was performed for each sampling, extended to two days when a dust-outbreak became evident. A total of two filters for each collection were processed for sequencing.

## 2.3 DNA extraction and Sequencing

DNA was extracted using the E.Z.N.A.® Soil DNA Kit (Omega Bio-Tek Inc.) as described by the manufacturer. Quality and quantity of the extracted nucleic acid were measured using a NanoDrop 2000 spectrophotometer (Thermo Fisher Scientific Inc.).

Amplification of the 16S-rRNA genes for sequencing was performed using the universal primers 27F-1492R (AGAGTTTGATYMTGGCTCAG and TACGGYTACCTTGTTACGACTT, respectively). PCR was carried out using Platinum® Taq High Fidelity DNA Polymerase (Life Technologies) in a PTC-200 Thermal Cycler (MJ Research Inc.) set as follows: 95°C for 5 min, (95°C for 0.5 min, 51°C for 0.5 min, 72°C for 2 min for 30 cycles), 72°C for 10 min and 4°C on hold. The amplification of the No Template Control (NTC) was negative. Next generation sequencing was carried out at the facilities of

the Porto Conte Ricerche Srl (Alghero, Italy). Briefly, amplicons were quality-checked on an agarose gel and purified using the Agencourt® Ampure® XP PCR Purification Kit. One ng of DNA was processed using the Nextera XT DNA Sample Preparation Kit (Illumina Inc.) and sequenced using the HiScanSQ (Illumina Inc.) with 93bp x 2 paired-end reads. Sequences were submitted to the European Nucleotide Archive(ENA) inside the "Dust Metagenome" BioProject with the accession numbers ERX836645-56.

**2.4 Data analysis**

Reads were cleaned on the basis of quality and fragments of Nextera adapters removed by Trimmomatic (Bolger et al. 2014) set at the value of 3 for leading and trailing trimming, and bases lower than 20 on a 4-base wide sliding window. Quality was confirmed by FastQC (http://www.bioinformatics.babraham.ac.uk/projects/fastqc/) and reads were analyzed with Qiime1.9.0 (Caporaso et al. 2010). The OTU table was created using the pick_otus script with the Closed-reference OTU picking strategy with the Greengenes reference OTUs database clustered at 97% (ver. gg_13_8). The same script checked against chimeric sequences using the Broad Microbiome Utilities' 16S Gold reference database (version microbiomeutil-r20110519). The OUT table was filtered based on the total observation count of an OTU at least of 3 and low abundance filtering of 0.005%. Finally, the OTU table was rarefied (subsampled) at 1109571 counts (equal to the sample with a lowest depth) for all the samples. Perl and the R-package Vegan were subsequently used for cladograms and distance-based clustering analyses, Ggplot, Plotrix and WindRose provided graphic support.
Molecular data regarding bacterial species compositional differences across different treatments were analysed by multivariate analyses (Principal Coordinate Analysis, PCoA; Principal Component Analysis, PCA, Discriminant Analysis of Principal Components DAPC),

and ecological indices calculation, using the Calypso online software tool (Zakrzewski et al., 2017). Prior to the analyses, the relative abundances of taxa were equalized by applying the total sum of squares scaling (TSS) normalization followed by square root transformation.

## 3. RESULTS

### 3.1 Meteorological events

To capture the air microbiota of Sardinia and to put in evidence taxa which could be associated to specific events (winds in northbound direction prone to carry dust from shores across the Mediterranean as opposed to calm air or slow flows from the opposite quadrant), wheather forecasts and other data on air circulation were regularly browsed to select suitable dates for the sampling. This allowed to integrate boundary conditions and environmental variables to assess possible correlations between these and microbial community fluctuations (Fiamma, 2016).

Dust-carrying air masses moved over the Mediterranean on May 21$^{st}$ through May 22$^{nd}$ 2014 towards north-east, covering the entire Sardinia island. PM10 concentrations throughout the second half of May at both North and South collection sites displayed increases in correspondence of the dust event (Supplementary Figure S1). Incoming dust of African origin was equally evidenced by charts reporting wind fronts and pressure (Supplementary Figure S2), and images from satellite (Supplementary Figure S3). The itinerary of particles was reconstructed by plotting 3-day backward trajectories of the air mass using a NOAA HYSPLIT model (Figure 2) which tracked the North-African zone as the source of the convective motion responsible for the dust discharge on Italy observed on May 21$^{st}$ -22$^{nd}$.

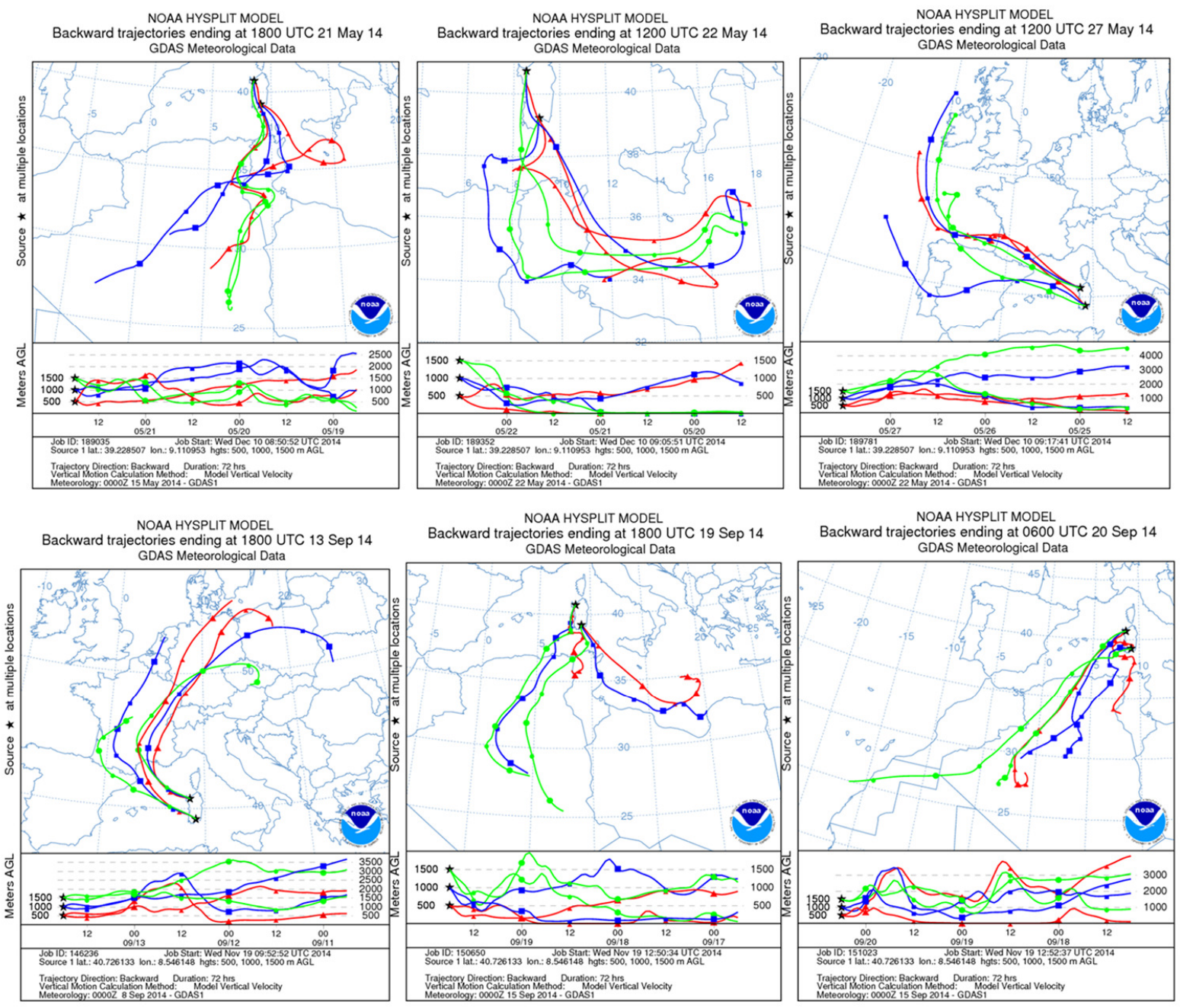

**Fig. 2. Upper panel**: 3-Day air mass backward trajectories calculated by the NOAA HYSPLIT model ending at 18:00 UTC May 21st, 12:00 UTC May 22nd and 12:00 UTC May 27th 2014 at both sampling sites. **Lower panel**: 3-Day air mass backward trajectories calculated as above, ending at 18:00 UTC September 13th, 18:00 UTC September 19th and 06:00 UTC September 20th 2014 at both sampling sites (credit to: ready.arl.noaa.gov/HYSPLIT.php).

May 27th was selected as "clear day" featuring a weather not conducive anymore for air convection from Africa to Sardinia. Such conditions consisted in overall European low pressures as opposed to high pressures over Mauritania, Mali, Libya and Algeria (Supplementary Figures S2 and S3). Particle back-tracking supported the evidence of a slow flow of air masses only from north-western corners on May 27th (Fig.2, upper panel, rightmost image).

An outbreak of dust on Sardinia was recorded again in 2014 during the second fortnight of September. Low pressure from the north-western coast of Spain to Morocco was opposed to parallel high-pressure system that extended over North Africa (Libya, Algeria, and Tunisia) through Sicily. This circumstance caused the flow of dust-carrying air masses over the Mediterranean basin, reaching in particular Southern Italy and Sardinia. Air movement from the African continent made air temperature rise to values above the usual September means, with a peak on Sep. 20th in Sassari (northern sampling site) and on Sep. 21st in Cagliari (southern sampling site) (Supplementary Figure S4-a,b). In relation to this condition, from September 19th through the 21st a dust outbreak from Sahara flew over the Mediterranean and entirely covered Sardinia. The relative wind fronts and pressure values are shown in Supplementary Fig. S5. Patterns of PM10 from daily records taken at both Sardinian sampling stations, also displayed a rise during the dust outbreak (Supplementary Fig. S4-c). Satellite imagery confirmed again the occurrence of incoming dust-loaded air masses from Northern Africa (Supplementary Fig. S6) consistent with their 3-day back-trajectories (Fig. 2 lower panel). Those confirmed that on September 19th - 20th air flows were from North African origin. About a week earlier instead, September 13th had featured low pressures on Italy while high pressures were recorded over the southern part of Morocco, Algeria and Mauritania. This picture was not permissive for any transport of air loads from Africa to Sardinia and the day was therefore considered as the "clear day" reference of the period. Air representative of the dust outbreak condition was thence sampled from Sep.19th through 20th, while the corresponding control air was collected on September 13th.

### 3.2 Bacterial community composition

To put in evidence microbial variation we envisaged the possibility of finding i) a local set of taxa, with specificity for one of the two sampling corners of the island, and a relative independence from the weather events, ii) those linked to the occurrence of dust outbreaks, (distinguishing in this case the first 12 hours timeframe from the second 12h one. iii) bacteria showing season-related fluctuations being specific or enriched in one or the other sampling times (May vs. September).

A synoptic view of the results at Phylum/Class level is shown in Fig. 3.

In terms of conserved taxa the core of those observed more regularly included classes as Gammaproteobacteria, Bacilli and orders as Actinomycetales; their maxima were seen in the May samples where those reached percentages above 90%, while their minima appeared in the Sassari controls in September with values around 50%. .

With respect to the Actinobacteria phylum, the Actinomycetales order was the one most commonly encountered, being found in all samples; in particular it was featured in the south-facing station (Cagliari), and its numbers tended to double in relation to the dust events. The overall levels of relative abundance as well as diversity within members of the Actinobacteria phylum increased from 5.66 % values, observed in May to 13 % in September. Particularly enriched were the Gaiellales and Solirubrobacterales order within the Thermooleophilia class. The orders within the Firmicutes phylum, that dominated the May samples, resulted Lactobacillales and Bacillales. Their relative abundances were higher in the Sassari (Notrhern Sardinia) control samples, than in those collected in the southern point of Cagliari, with values of 37 % vs. 12 % respectively. At the same time an unchanging level of 25% was recorded in both control and dust samples in the south-facing location. In the September samples the situation was different as in both controls those orders were below 15%, while during the dust outbreak it was 25 % in both Cagliari and Sassari stations.

The May sampling was also characterized by a large share of Gammaproteobacteria, a class reaching 75% of the dust-related spring samples in Sassari. In particular Pseudomonadales and Enterobacteriales were constantly observed. Some taxa constituting spring-signature cases were detected in the Alteromonadales with *Marinimicrobium*, *Marinobacter* and taxon

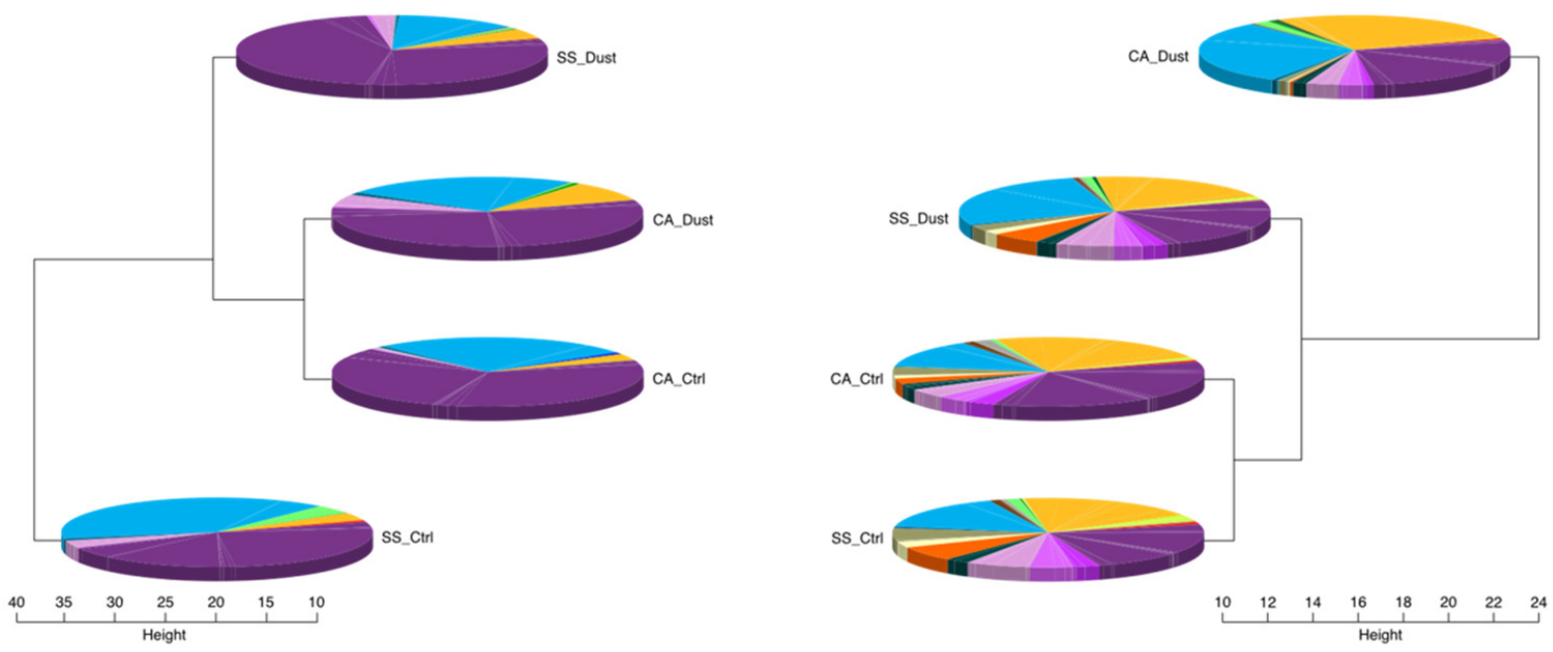

**Fig. 3**: Cluster Dendrogram (Euclidean distance method, complete linkage) on the identified bacterial orders. May samples (the four left column pie charts) compared sideways to the corresponding September samples (right column pie charts). Data from the two sampling stations of Sassari (SS, northern Sardinia) and Cagliari (CA, Southern Sardinia) are shown, comparing the two wind regimes ('Dust': during dust outbreaks under winds from Africa, and 'Ctrl': Control, under winds from Europe). Pie colours coding (clockwise): yellow: Actinobacteria; light green: Acidobacteria; red: Verrucomicrobia; dark purple: Gammaproteobacteria; fuchsia: Deltaproteobacteria; light fuchsia: Betaproteobacteria; light pink: Alphaproteobacteria; black: Planctomycetes; orange: Nitrospirae; white: NC10; khaki: Gemmatimonadetes; blue: Firmicutes; brown: Chloroflexi; grey: Chlorobi; green: Bacteroidetes; dark green: Armatimonadetes.

OM-60. For the fall period instead, Xantomonadales recurred with some genera in the Sinobacteraceae family, amounting from 1.2% to 2% respectively in the dust-free controls of Cagliari and Sassari,

Within the Alphaproteobacteria class, Caulobacterales, with genera related to *Brevundimonas* were at 1% relative abundance level in the Cagliari samples collected during the dust episode of May.

The Rhizobiales order was present in both seasons with a 3% peak in spring (Cagliari, dust-related), dropping to 1.5 % in all fall analyses. In the same period Rhodospirillales showed a relative increase, particularly in the controls in Sassari where they reached 3.4%.

The Burkholderiales (Class Betaproteobacteria) of the population were found at 1 % in May within the dust-related sequences and at higher values, reaching 2.7% in Cagliari and 4% in Sassari, in the controls of September.

Some groups appeared rather season specific as the Mollicutes for May, while the Pirellulales order (in the Planctomycetes phylum) and the classes of Nitrospira and Gemmatimonadetes characterized the September sampling.

To better refine the bacterial deposition dynamics during the outbreaks, during the total 24h sampling time, two sampling sub-periods were set, splitting the total collecting span into two 12-h lapses, by changing the filters after the first one and separating the collected material as different samples. An increase in the inflow of air particulate was observed for the 12-24 h period.

This set up was also functional to individuate taxa that would display high variation in relation to dust events in comparison to those who would not. The latter were considered to represent the common core of bacteria that were constantly present in samples, irrespective of the changing meteorological events. To apply this distinction, the criterion was to set a cutoff value with respect to the percent of variation occurring between the first 12 h of the collection time and the second half of it. Only the taxa which displayed a mean variation higher than ½ of the corresponding standard deviation were considered. The resulting level of variation in the two sampling stations is reported in Tab. 1 and the corresponding number of orders is displayed in

| Sample | Avg. variation % | Min variation % | Max variation % |
|---|---|---|---|
| Sassari May - Dust | 1.4 | 0.05 | 6.7 |
| Cagliari May - Dust | 2.1 | 0.5 | 5.0 |
| Sassari Sepember - Dust | 1.3 | 0.4 | 5.4 |
| Cagliari September - Dust | 4.7 | 1.1 | 11.4 |

**Tab. 1:** Average, minimum and maximum percent variation between taxa counts harvested in the first 12 hours sampling period of the dust event and those harvested in the subsequent 12 hours sampling period. Only taxa displaying a difference in percentages higher than half of their standard deviation were selected for the present comparison.

| Site and period | Total Orders | Selected Orders | % of total orders |
|---|---|---|---|
| Sassari May | 56 | 16 | 28% |
| Sassari September | 103 | 28 | 28% |
| Cagliari May | 52 | 11 | 21% |
| Cagliari September | 87 | 14 | 16% |

**Tab 2.** Community diversity at order level of taxa occurring during dust events and of those displaying variations higher than half the standard deviation between the first 12h and the second 12h sampling period (selected orders). The percentage of orders selected upon this criterion over the total of the orders observed in samples collected during the dust events is indicated.

Tab. 2. The Sassari (North-facing) collection site was the one that in both seasons resulted to feature the highest number of significantly changing taxa. The identities of these are shown in Supplementary Fig.S7 (May event), and Supplementary Fig. S8 (September event). In the graphs, the first 12h lapse is plotted above the baseline and the second (12-24 h) is on the specular position below.

As regards the ecological indexes characterizing the communities, species diversity and evenness values were calculated, and results are shown in Tab. 3. The difference that can be appreciated is mainly relative to the series of September samples, in which all had higher values for each of the indexes when compared to the May ones. Conversely, neither the presence of dust events nor the sampling location appeared to confer relevant differences in this respect.

The numerical effect of the different sampling season on bacterial communities is visible in Tab. 4, comparing the mean relative abundances of the main orders in the two sampling months, grouped independently from site and meteorology events. Among the most evident phenomena. the September campaign shows the enrichment in the Actinomycetales order and in a number of others that were below detection in the May sampling. In parallel, the diminution of the formerly dominant Enterobacteriales and Pseudomonadales, and the substantial stability of the Bacilli across the compared times were observed.

The patterns of conservation and diversity involving the bacterial communities analyzed were subsequently inspected by multivariate approaches. Principal Component Analysis yielded an output (Fig. 4) that confirms how a separation of communities can be viewed only when considering the seasonal factor (Fig.4. A), while the variables of dust vs. calm air, or the sampling location, led to plots with heavily overlapping patterns. The May vs. September divide occurs along the horizontal axis, i.e. the one explaining the highest fraction of variation (35%). The same phenomenon is reproduced with a higher support (54 %) in a parallel

| Month, Event, Place | Simpson 1-D | Shannon H | Evenness |
|---|---|---|---|
| May Dust SS h 1-12 | 0.771 | 2.062 | 0.151 |
| May Dust SS h 12-24 | 0.740 | 1.902 | 0.156 |
| May Dust CA h 1-12 | 0.833 | 2.175 | 0.183 |
| May Dust CA h 12-24 | 0.833 | 2.205 | 0.197 |
| May Ctrl SS | 0.794 | 2.064 | 0.164 |
| May Ctrl CA | 0.778 | 1.900 | 0.142 |
| Sep. Dust SS h 1-12 | 0.928 | 3.187 | 0.260 |
| Sep. Dust SS h 12-24 | 0.914 | 3.015 | 0.240 |
| Sep. Dust CA h 1-12 | 0.887 | 2.792 | 0.212 |
| Sep. Dust CA h 12-24 | 0.838 | 2.339 | 0.176 |
| Sep. Ctrl SS | 0.948 | 3.438 | 0.311 |
| Sep. Ctrl CA | 0.936 | 3.292 | 0.286 |
| May: mean ±SD | 0.79 ± 0.04 | 2.05 ± 0.13 | 0.17 ± 0.02 |
| September : mean ±SD | 0.91 ± 0.04 | 3.01 ± 0.40 | 0.25 ± 0.05 |

**Tab 3**. Ecological diversity and evenness indices resulting from the sequence checklist analysis in the different samplings.

| Phylum | Class | Order | Mean percentage May | Mean percentage September |
|---|---|---|---|---|
| Proteobacteria | Gammaproteobacteria | Enterobacteriales | 27.40 | 11.55 |
| Proteobacteria | Gammaproteobacteria | Pseudomonadales | 26.67 | 9.90 |
| Firmicutes | Bacilli | Lactobacillales | 18.67 | 15.96 |
| Actinobacteria | Actinobacteria | Actinomycetales | 5.66 | 13.36 |
| Firmicutes | Bacilli | Bacillales | 4.46 | 6.56 |
| Proteobacteria | Gammaproteobacteria | Alteromonadales | 2.77 | 0.76 |
| Proteobacteria | Gammaproteobacteria | Xanthomonadales | 1.51 | 2.63 |
| Proteobacteria | Gammaproteobacteria | Aeromonadales | 1.51 | 0.96 |
| Proteobacteria | Alphaproteobacteria | Rhizobiales | 1.35 | 1.59 |
| Bacteroidetes | Sphingobacteria | Sphingobacteriales | 1.00 | 1.04 |
| Proteobacteria | Alphaproteobacteria | Rhodospirillales | 0.06 | 2.02 |
| Actinobacteria | Acidimicrobiia | Acidimicrobiales | 0.05 | 2.01 |
| Nitrospirae | Nitrospira | Nitrospirales | 0.01 | 3.38 |
| Actinobacteria | Thermoleophilia | Gaiellales | 0.00 | 2.81 |
| Actinobacteria | Thermoleophilia | Solirubrobacterales | 0.00 | 2.37 |
| Gemmatimonadetes | Gemm-1 | Gemm-1 | 0.00 | 2.52 |

**Tab. 4**. Percent frequency of sequences belonging to the indicated orders in the averaged data of all samplings (Dust and control) of each seasonal sampling period (May or September). Data in which frequencies were higher than 1% in at least one of the two seasons are reported. These represent the 91.1% of the total sequences for the May sampling (on a total of 65 orders found) and 79.4% of the September sampling (on a total of 118 orders found).

ordination approach, the principal Coordinate Analysis (Fig. 5. A). In the same figure the main differences occurring in community structure between the two sampling times are further explored by reporting the ecological indexes of Shannon species diversity and community evenness resulting from grouping the data and separating them only in relation to the sampling period variable, irrespective of meteorology events and collection sites. The superiority of the September values in both indexes, and particularly for the taxa diversity, is supported by the significance of the p values of discrimination between samples thereby reported.

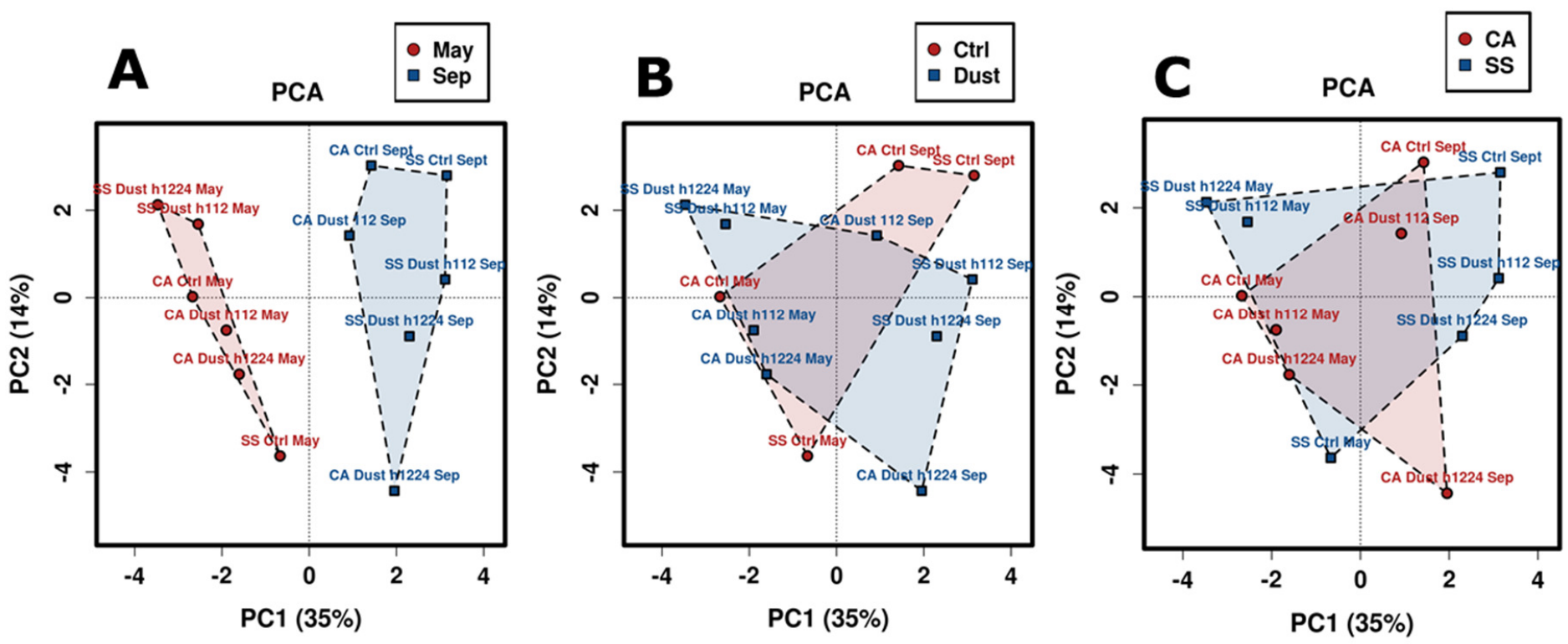

**Fig. 4. Principal Component Analysis (PCA) ordination plot of the bacterial community compositional data.** Polygons encompassing the positions of three different variables are drawn to visualize season **(A)**, ongoing meteorological event **(B)**, or sampling location **(C).**

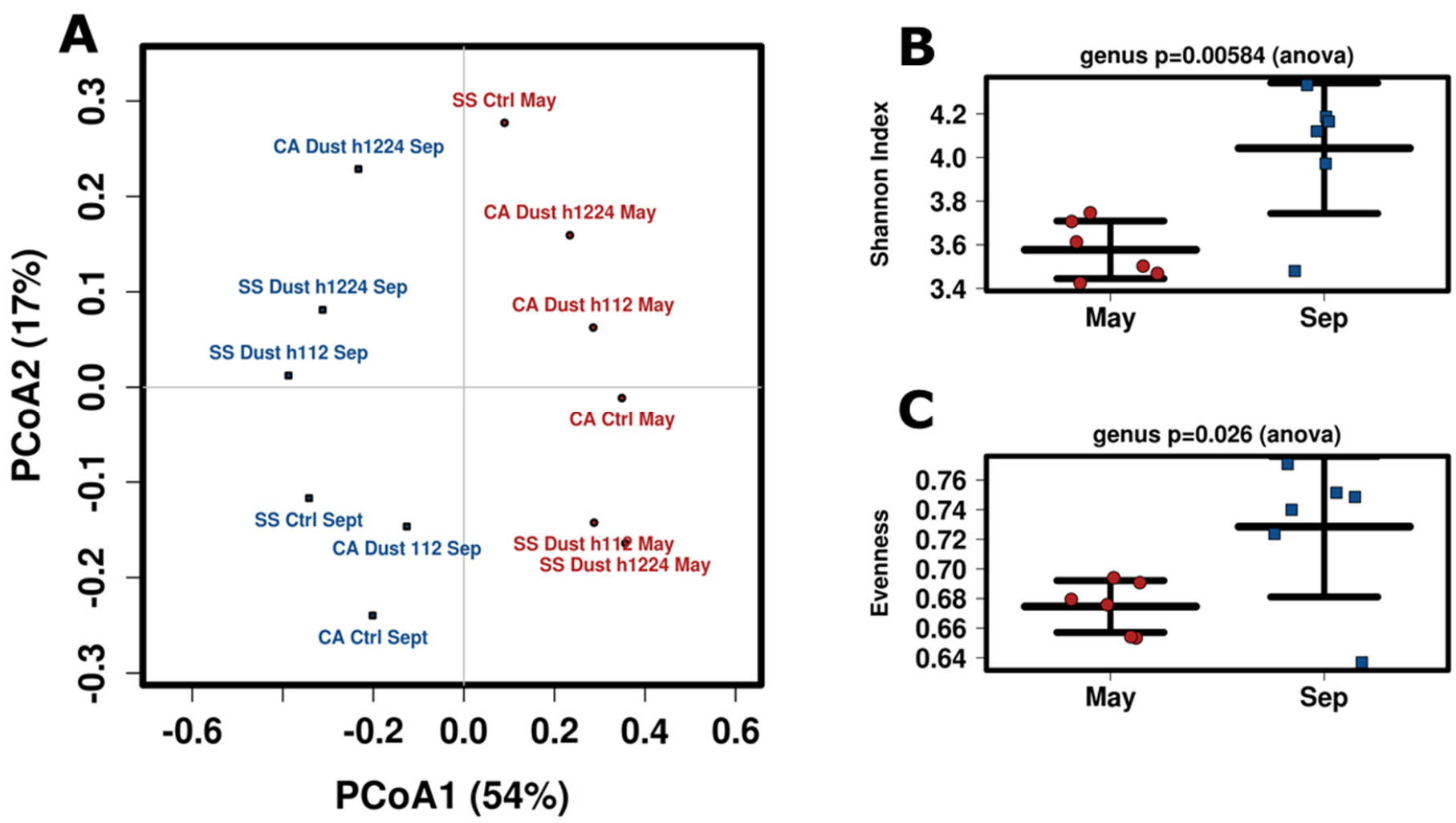

**Fig. 5. A. Principal Coordinate Analysis.** Dataset is as in Fig. 4. **B.** Shannon index of species diversity boxplot comparison between the two season's samplings (source for calculation: square root of total sum of squares data transformation). **C.** Community evenness index comparison on the same data. The significance of differences by ANOVA is reported over each diagram.

The higher strength of clustering of the sampling date groups with respect to the alternative ones (meteorological or geographical) was verified by running a Discriminant Analysis on the Principal Component ordination (DAPC) in which the data are first transformed by PCA, from which, clusters are subsequently identified using Discriminant Analysis, thus partitioning sample variance into the between-group and within-group components. Results are shown in Fig. 6. Besides confirming the sampling season as the strongest driver of community change, the analysis further shows that the dust vs. control clustering is acting more efficiently than the Sassari vs, Cagliari sampling site comparison. This allows to draw a hierarchical ranking of the variables in shaping the bacterial airborne communities, in which, noting also the different

scale of the horizontal axis (Discriminant function 1) adopted for the three graphs, the order results : Season >> Meteorology > Geography.

In order to determine which bacterial taxa were mostly accompanying/causing those changes in a statistically significant manner, and to rank their individual importance in this phenomenon, we run an analysis of the differentially featured taxa, testing both an ANOVA variance analysis and a non parametric Wilcoxon Rank test verification of the ranking. The two tools gave coherent scores and the results of the ANOVA output are shown in Supplementary Table S1. A total of 76 taxa were found featuring p values < 0.05, from which, upon applying a stringent Bonferroni-adjusted p value correction, six of those stood above the significance cutoff, and all within minimal false discovery rate values ( FDR < 0.005). All of them were cases which were highly reduced in September in comparison to May. The taxa included as the most effective in explaining the differences (p value = 0.000019, the order Oceanospirillales, known as marine oil spill-associated bacteria (Cao et al, 2013), followed by known animal parasites as the Coxiellaceae family (Lory, 2014), marine extremophyles as the Thiohalorhabdales (Tian et al., 2017), and three species of *Pseudomonas*, including the pathogenic *P. viridiflava* (Hu et al., 1998), the decontamination-associated *P. nitritireducens* (Wang et al., 2012) and *P. alcaligenes* which is reported also a human pathogen (Suzuki et al, 2013). The two corresponding analyses of differentially represented taxa by meteorology or by geography, i.e., grouping dust vs. calm air or Cagliari vs. Sassari sites did not yield any significantly supported cases under the Bonferroni-adjusted p values stringent condition (data not shown).

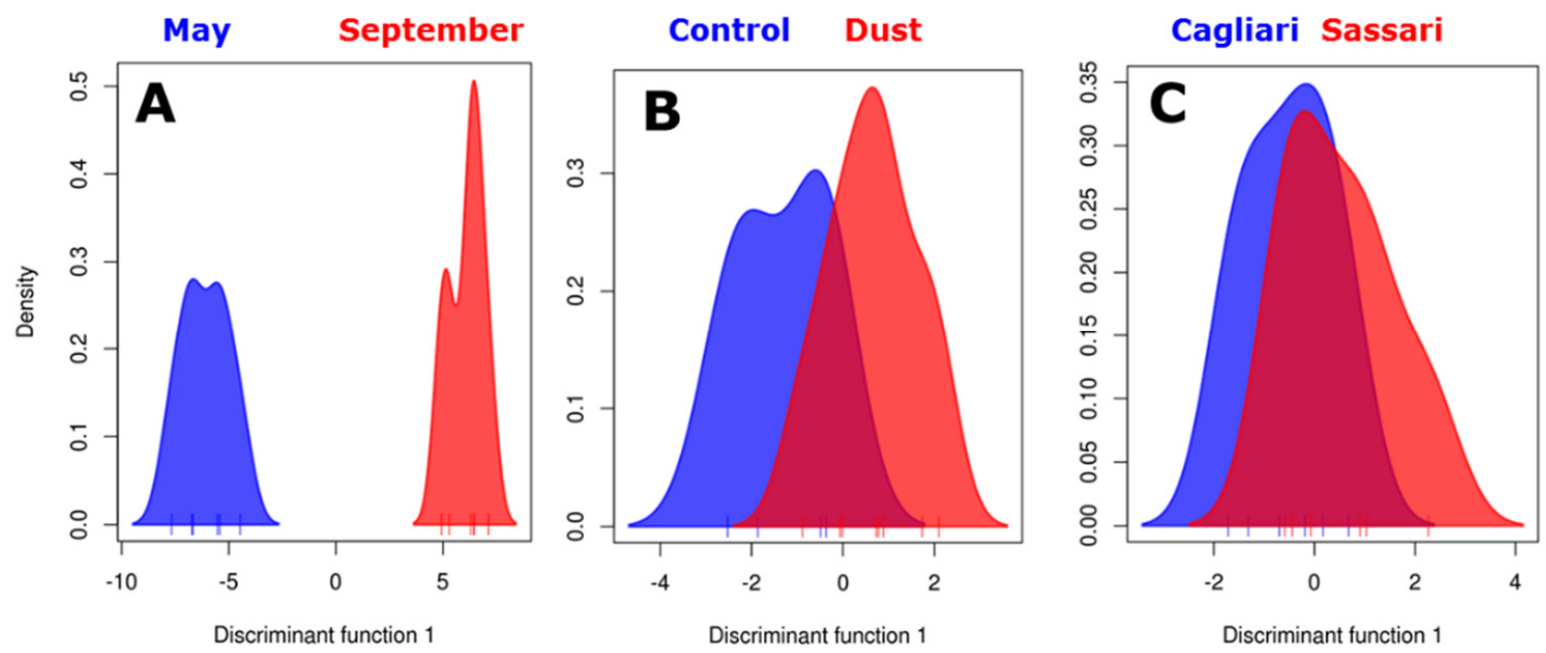

**Fig. 6. Discriminant Analysis of Principal Components analysis**. Group partitioning involved **A**: season; **B**: ongoing meteorological event; **C**: sampling location.

In order to compare all communities with each other and extract further information on their degrees of divergence, the sequencing data were analyzed by individual comparisons across sites and dates. The results of each of the 66 pairwise combinations are shown in Tab. 5, displaying the Bray Curtis similarity values between each couple of communities. Color-based conditional formatting applied to the values allows to appreciate how all the comparisons involving different seasons show the most divergent scores (red shades) in comparison to those within the same season, that show much more similarity, with few exceptions related to dust events and depending on the aspect faced by the collecting site with respect to the incoming wind direction.

| Different season | | | | | |
|---|---|---|---|---|---|
| *Different place* | | | *Same place* | | |
| SS D1-12 May | CA D1-12 Sep | 0,476 | SS D1-12 May | SS D1-12 Sep | 0,348 |
| SS D1-12 May | CA D12-24 Sep | 0,344 | SS D1-12 May | SS D12-24 Sep | 0,380 |
| SS D1-12 May | CA Ctrl Sep | 0,449 | SS D1-12 May | SS Ctrl Sept | 0,355 |
| SS D12-24 May | CA D1-12 Sep | 0,416 | SS D12-24 May | SS D1-12 Sep | 0,317 |
| SS D12-24 May | CA D12-24 Sep | 0,321 | SS D12-24 May | SS D12-24 Sep | 0,372 |
| SS D12-24 May | CA Ctrl Sep | 0,423 | SS D12-24 May | SS Ctrl Sep | 0,312 |
| CA D1-12 May | SS D1-12 Sep | 0,471 | SS Ctrl May | SS D1-12 Sep | 0,450 |
| CA D1-12 May | SS D12-24 Sep | 0,532 | SS Ctrl May | SS D12-24 Sep | 0,507 |
| CA D1-12 May | SS Ctrl Sep | 0,388 | SS Ctrl May | SS Ctrl Sept | 0,420 |
| CA D12-24May | SS D1-12 Sep | 0,481 | CA D1-12 May | CA D 1-12 Sep | 0,536 |
| CA D12-24May | SS D12-24 Sep | 0,538 | CA D1-12 May | CA D12-24 Sep | 0,476 |
| CA D12-24May | SS Ctrl Sep | 0,417 | CA D1-12 May | CA Ctrl Sep | 0,526 |
| SS Ctrl May | CA D 1-12 Sep | 0,448 | CA D12-24 May | CA D 1-12 Sep | 0,530 |
| SS Ctrl May | CA D12-24 Sep | 0,523 | CA D12-24 May | CA D12-24 Sep | 0,479 |
| SS Ctrl May | CA Ctrl Sep | 0,469 | CA D12-24 May | CA Ctrl Sep | 0,525 |
| CA Ctrl May | SS D1-12 Sep | 0,387 | CA Ctrl May | CA D 1-12 Sep | 0,450 |
| CA Ctrl May | SS D12-24 Sep | 0,464 | CA Ctrl May | CA D12-24 Sep | 0,416 |
| CA Ctrl May | SS Ctrl Sep | 0,355 | CA Ctrl May | CA Ctrl Sep | 0,453 |
| | | | | | |
| **Same season (May) Control taken after dust** | | | | | |
| *Different place* | | | *Same place* | | |
| SS D1-12 May | CA D1-12 May | 0,649 | SS D1-12 May | SS D12-24 May | 0,736 |
| SS D1-12 May | CA D12-24 May | 0,627 | SS D1-12 May | SS Ctrl May | 0,545 |
| SS D1-12 May | CA Ctrl May | 0,640 | SS D12-24May | SS Ctrl May | 0,522 |
| SS D12-24 May | CA D1-12 May | 0,709 | CA D1-12 May | CA D12-24May | 0,802 |
| SS D12-24 May | CA D12-24 May | 0,655 | CA D1-12 May | CA Ctrl May | 0,784 |
| SS D12-24 May | CA Ctrl May | 0,713 | CA D12-24 May | CA Ctrl May | 0,791 |
| CA D1-12 May | SS Ctrl May | 0,618 | | | |
| CA D12-24 May | SS Ctrl May | 0,679 | | | |
| SS Ctrl May | CA Ctrl May | 0,676 | | | |
| | | | | | |
| **Same season (September) Control taken before dust** | | | | | |
| *Different place* | | | *Same place* | | |
| SS D1-12 Sep | CA D 1-12 Sep | 0,600 | SS D1-12 Sep | SS D12-24 Sep | 0,718 |
| SS D1-12 Sep | CA D12-24 Sep | 0,604 | SS D1-12 Sep | SS Ctrl Sep | 0,705 |
| SS D1-12 Sep | CA Ctrl Sept | 0,656 | SS D12-24 Sep | SS Ctrl Sep | 0,616 |
| SS D12-24 Sep | CA D 1-12 Sep | 0,599 | CA D 1-12 Sep | CA D12-24 Sep | 0,554 |
| SS D12-24 Sep | CA D12-24 Sep | 0,655 | CA D 1-12 Sep | CA Ctrl Sep | 0,589 |
| SS D12-24 Sep | CA Ctrl Sep | 0,656 | CA D12-24 Sep | CA Ctrl Sep | 0,441 |
| CA D 1-12 Sep | SS Ctrl Sep | 0,523 | | | |
| CA D12-24 Sep | SS Ctrl Sep | 0,434 | | | |
| SS Ctrl Sep | CA Ctrl Sep | 0,705 | | | |

**Table 5. Bray Curtis similarity values between the bacterial communities composition resulting from pairwise comparisons of all samples.** Abbreviations: SS: Sassari; CA:

Cagliari; D1-12: dust event, first 12 hour period; D12-24: dust event, second 12 hour period; Ctrl: control conditions (absence of dust events); Sep: September.

## 4. DISCUSSION

In the present study the filtered air particulate was analyzed in different seasons and under different wind regimes, using culture-independent DNA sequencing-based approaches targeting the species-diagnostic 16S-rRNA genes from the air-carried bacterial community and an Illumina next generation sequencing platform. Sites were selected also because of their opposite positions facing Africa (Cagliari) or continental Europe (Sassari). The analysis was performed within a 7-month time lapse, March to September, chosen also as it offers higher probabilities of weather shifts favoring both northern- and southern-winds (Israelevich *et al.* 2012). This timeframe proved suitable to the scope as it was possible to exploit two episodes in which dust outbreaks carried by winds of African origin occurred and were preceded and followed by inversions of the air circulation offering control sampling periods with opposite features.

The central goal of this study was to assess which variables (sampling time of the year, dust outbreak vs. calm atmosphere, and north-facing vs. south-facing collection site) would be most effective in determining airborne community divergence or homogenization.

One first general aspect that can be commented is a higher diversity of the communities during the September sampling in comparison to May, independently from the dust events and from the sampling station location.

This phenomenon, besides the ecological values differences (Tab. 2, Tab. 3, Fig.5, and Tab. S1) can be also appreciated visually, by comparing the left and the right pie charts in Fig.3,

(featuring community composition at order-rank level, and the corresponding cluster analysis based on their relative percentages), and noticing the more complex color-coded pattern of the latter sampling, showing also a consistent similarity of most color sectors presence and proportions. It is not possible from these single-year data to deduce whether such increase could be part of a recurring seasonal phenomenon causing, cyclically, higher species richness after summer periods, or if what we observe could be part of a different pattern of stochastic variability.

Nevertheless the overall partitions of systematic groups observed in a given sampling time, irrespective of dust outbreaks or sampling corner of Sardinia, share much more similarity within the samples of that period than with any of those collected in the other season. It appears that in general, air collected during dust discharge from a Saharian wind can account for less variation over its reference control sampling than the choice of sampling that site four months apart.

In our prior work (Rosselli et al., 2015) we had studied community composition in the same Sardinian stations in a short period of winter (in late February) during and after a single dust-carrying event. In that study the main feature evidenced was the existence of a conserved core microbiome, encompassing 86-95 % of the taxa, to which the incoming dust would cause some detectable diversity variation but on a rather limited proportional scale. Such minor effect of the dust-lifting storms observed in winter is in fact confirmed in the present work in which the time of the year factor appears as the variable of major order in shaping community structure and richness.

Literature reports have pointed out differences in airborne microbial composition between seasons; peaks of fungi causing invasive infections in humans were signaled in spring whereas higher proportions of allergenic fungi were observed in fall (Yamamoto et al. 2012).

Consistent with the present data a higher diversity of both fungal and bacterial airborne cells in late summer and early fall has been observed in United States-based surveys (Bowers et al. 2012¸ Bowers et al., 2013).

Hypotheses to explain the increase in circulating taxa widely observed in the fall sampling campaign can be formulated. In first instance one should consider whether there could have been a change in the prevailing winds origin or direction across the period that encompasses the two sampling seasons. This can be evaluated upon inspecting publically available meteorology records showing the wind roses for the two sampled localities. These data, from March to November, for the Cagliari and Sassari weather stations, are shown in Supplementary figures Fig. S9 and Fig S12, respectively. In the Cagliari plots (southern Sardinia) it can be observed that between May and September there was basically no variation of the wind patterns, with the prevailing ones blowing towards North-West, with stable intensities. Likewise in the Sassari area (Fig. S12), although some fluctuations in the strength of the westbound winds can be seen, the dominant air motion throughout the period remains the one heading South. In essence these data allow to rule out that the change in community patterns could be due to major air-driven events of taxa immigration from other insular or continental sources.

In addition to the wind orientation and force, data from the two stations regarding temperature and humidity of the same winds can be analyzed (Supplementary Fig. S10, Fig. S11, Fig. S13, Fig. S14). Humidity values from May to September winds tend to be rather similar, whereas air temperatures increase in line with the summer progression. These data do not account by themselves for events of species enrichment either.

Another aspect that can be verified is to compare the two periods in terms of PM10 particulate concentration; these are reported in Supplementary Fig. S1.C (May) and Fig. S4.C (September). Although there are obvious peaks of PM10 in correspondence with the dust

outbreaks dates, the basal levels of PM10 concentrations before and after those, are rather similar in the spring and fall period. This rules out the possibility of a diversity rise as linked to a general increase of such small particles trafficking over the areas.

The observed data reveal that, while dust-associated winds can account for some specific limited ingression of taxa, a far more noticeable pattern appears consisting in a successional rise of taxa diversity. It is not yet possible to establish whether this occurrence could be linked to late summer in relation to the climatic conditions of the season. The second part of the summer, especially in the Mediterranean regions, is characterized by prolonged drought alternated to irregular thunderstorms. The income of a thunderstorm is accompanied by convective instability of the atmosphere and this phenomenon has been already pointed out as conducive to the emission and transport of fungal spores plumes (Burch and Levetin, 2002). A possible explanation for a richer pattern of airborne microbes after several weeks of prevailingly dry climate can be sought in the acknowledged fact that those seasonal conditions enhance the daytime height of the planetary boundary layer over Europe and continental US (Seidel et al. 2012), and that the ensuing low pressures foster the turbulence near ground and the overall convection, resulting in a frequent uplift of particles from land surfaces. In addition, it could also be postulated that the dryer and warmer summer conditions can eventually lead to partial cell dehydration in microbes lying at soil or vegetation surface, resulting in lighter cell weights more prone to be advantageously lifted by the local low layers air turbulence.

A further factor that can be hypothesized to have played a role in reducing the diversity of airborne community samples in May, comes from the analysis of the differentially featured taxa between the spring and the fall samplings (Tab. S1) where the strongest statistically significant differences were six taxa that resulted highly enriched in the former period and that, as cited above, included marine bacteria associated to oil spill-related oleovory phenotypes, extremophyles, and potential pathogens. These occurrences can be interpreted as possible clues

for a transient event of water pollution around the sampled areas that could have impacted also on the overall airlifted microbial diversity.

In addition to the above, a series of considerations can be drawn upon inspecting the pairwise community difference analysis, whose similarity values are shown in Tab. 5. It also needs to be recalled that, in order to examine the effect of a dust-free period, in May the control (May 27$^{th}$) was sampled after the dust event (May 21$^{st}$ – 22$^{nd}$), while in September the control (Sep 13$^{th}$) was taken before the new dust outbreak (Sep 19$^{th}$ – 20$^{th}$). Therefore, the summer, within which communities could undergo dust-independent changes, is in fact framed between the two control points, chosen as representative of dust-free atmosphere following a wind direction reversal. Within those months there were no dust-carrying wind outbreaks from the African land. This enabled also to verify whether a relatively long period without dust intrusions could have allowed an overall homogenization of the bacterial airborne communities over the island Sardinia.

The first consideration that stems from the global view of these data is once again that the most distant communities are those compared from different seasons (Tab.5, upper section) as evidenced by the red-to-yellow shades of the conditional formatting. It is worth noticing in this respect that no particular difference appears when comparing communities between those collected from the same site (right panel in the upper section) or in the cross-comparison between the two different places. Moreover, in these samplings from different seasons, the effects of the dust events in comparison to calm air with dust-free wind regimes, is not apparent, being diluted in the major season-related divergence of the communities.

When inspecting cases of the same season, the situation in May is representing a comparison picturing the recovery after the dust event, as the control follows the outbreak. One evident aspect in that is how the juxtapositions within the same place, feature the most similar cases (darkest shades of green) with the notable exceptions linked to the dust outbreak in Sassari,

which is the North-facing station (notice the two yellow-shaded values). On the contrary, such divergence does not appear at all in the South-facing Cagliari site. One interpretation of this interesting difference is that in the May control, when the air flux reversed after the dust event, the wind blowing from the northern quadrant, was conveying in Sassari air masses that came straight from the sea; while on the opposite corner (Cagliari) instead, the same air had passed over the whole Sardinia. Thus, the Northern collection station received sea-sweeping air, bringing 'fresh' taxa, i.e. not belonging to the Sardinian land-related common bacterial core of the season, while the southern station of Cagliari received instead land-sweeping air that had travelled all the way over the island latitudinal extension, and that therefore would have become mixed with the island-related core of biota. Thus, the sea-related entries would bring little contribution to the southern site communities after more than 100 miles of travelling and being diluted through the terra firma atmosphere. This would explain why, in May, shifting from dust outbreak to control in the Southern location, did not bring community divergence as it did in the Northern one.

The imprint of the common Sardinian core on homogenizing communities when the dust preceded the control, is also testified by the left panel (same month of May but different places) resulting in all green shades of medium value, showing that in such situation there was little difference also between different places.

An independent and indirect confirm of this interpretation is given by the situation in September. In that case, the African dust-carrying event was set to be taken after the control; this originated a reversed situation in comparison to the one observed in May; this time the place where the dust-outbreak did not bring particular change was the Northern site, Sassari, as the northbound wind from African origin had supposedly already discharged its load while passing over the land of the island from which, at the same time it would have lifted a vast portion of land-related common biota. Vice versa, in Cagliari, appreciable changes occurred

in relation to the dust arrival, which support the view of air blown over the sea plus dust, as the elements causing changes due to the new kinds of bacteria that hit this side at the frontal south-facing port of entry of the island. The left panel of the section (same season, September, different place) further confirms this as: (a) the strongest drivers of community divergence (yellow to orange colors) are flagged by the two comparisons between the Sassari control and the Cagliari dust situations, and the second of those, in the 12-24 hours window of the dust event is progressively more divergent then that recorded during the first 12 h (0.434 vs. 0.523 similarity value). Moreover, the comparison between the two sites in the September control before dust, shows a rather high similarity (0.705), that is the highest among the September comparisons of different sites, which confirms that, before the dust outbreak, when both localities had experienced a long period devoid such phenomena, the two places had achieved a high degree of uniformity in spite of their distance. In that status, both communities were also profoundly different from their composition in May. A period of over 100 days without intrusions of dust-carrying northbound winds, appears to have accompanied an appreciable concerted successional change of the air-associated bacteria upon the Sardinian territory.

Essentially it appears that when airborne dust has to cross longitudinally the entire large island, it reaches the Northern sampling site (Sassari) less charged with community-changing potential, and/or, that it must have lifted bacteria from of the Sardinian common core, thus causing little variation upon their discharge over a station on the same island. On the contrary, when landing on the south-facing outpost of Cagliari, coming straight form Africa and, until that moment, having travelled over the sea only with air-lifted transcontinental dust, those air masses delivered in the south outpost of Cagliari an appreciably novel community. The geographic position of the sampling sites in relation to the wind origin appears therefore to play a major role in the patterns outcome. This supports the view that, in case of dust outbreaks, Cagliari, in the south, is at the forefront of changes that are substantially attenuated before they

reach Sassari; and vice versa, in case of reversed winds. A distance of >100 miles appears sufficient to absorb and buffer wind-borne taxa immigration in quantitative terms, from either side.

In conclusion, data are supportive of season related successional phenomena, involving a pattern of diffuse contemporary colonization over large portions of land, whose effect in shaping and homogenizing communities is stronger than the one conferred by occasional transcontinental discharges.
These clues entail novel aspects for our better understanding of microbial transport and spread across territories, of the epidemiological patterns for clinically relevant taxa, and can foster the predictive modeling of overall environmental microbiology dynamics.

**Supplement**.

The supplement related to this article is available online at: https……

**Author contributions**.

PC conceived the project; RM, AS, AC, and PD designed the experiments and supervised the project. RR, MF, MD, GP1 and GP2 carried out the analyses and interpreted the data. AS wrote the manuscript.

**Competing interests**

The authors declare that they have no conflict of interest.

**Acknowledgments**

This work was supported by a grant from Regione Autonoma della Sardegna, Legge 7/2007, Project D.U.S.T. (Desert Upon Sardinian Territory), CRP-17664.

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
