# Peer review of "Determining the hierarchical order by which the variables of sampling period, dust outbreaks occurrence, and sampling location, can shape the airborne bacterial communities in the Mediterranean basin"

_Biogeosciences, 2020_

## Referee Comment (RC1) · Anonymous Referee #2 · 23 Dec 2020

**Review bg-2020-324 :**

.

**General comment**

Rosselli et al. (2020) present a paper entitled «Determining the hierarchical order by which the variables of sampling season, dust 2 outbreaks occurrence, and sampling location, can shape the airborne bacterial communities in the Mediterranean basin».

The presented results are of great interest as there is an urgent need of data about bioaerosol's biodiversity and transport together with the understanding of the parameters shaping the air microbiome

The case studies presented in this paper are very well chosen as they consider different situations, all taken in Sardinia with sampling on two different sites: Sassari (North Sardinia) exposed to European influence, Calgari (South Sardinia) exposed to African influence. Also two time periods have been considered: May and September, as well as dust and clear sky events. NGS-based taxonomic analysis has been carried out in all these samples and compared using various pertinent statistical tools.
The results allowed to propose a hierarchy of the variables determining the biodiversity of the collected bioaerosols: sampling season >> ongoing meteorological events > sampling site. The sampling period had clearly the major impact.

This paper is worth publishing in Biogeosciences when the authors answer some major remarks and possibly make new complementary analyses.

**Major concerns**

My major concern lies on the fact that the authors did not compare in an extensive way the results obtained in this study with those previously published by Rosselli et al (Sci Rep. 2015). In my opinion they should have included these first results obtained at the same location sites (Calagari and Sassari), under dust and clear sky conditions but at a different season (February). The same data sets recorded by the same authors are available and could be compared directly with the present one using the same statistical tools.

This comparison would bring strong arguments to generalize the hierarchy of the variables influencing the structure of the microbial communities and make the conclusions clearer.

The discussion presented in the present paper has to be consistent with the one published earlier to be really acceptable and sound.

For instance, the present paper (Rosselli et al. 2020, Table 3, Page 17) clearly shows that dust events have little influence of the biodiversity indexes (Simpson 1-D, Shannon H), this is quite contradictory with what is claimed in Rosselli et al. 2015 (Table 2, Page 4). Also this dust influence is illustrated in PCA and Cluster dendrograms presented Figure 4 (Page 5) and 5 (Page 6) of the results published in 2015 (Rosselli et al.), again this is contradictory with figures 4, 5 and 6 (Pages 18, 19 and 21 respectively) of the present manuscript. Could the authors comment on these results and possibly merge the data of the two papers with new statistical analyses integrating all the data. It is quite important to clarify the influence of not of dust events on the microbiome compositions.

Another point concerns the evidence of a "major conserved core microbiome" that could be considered as a "global Sardinian air microbiome" ( Figure 3 Page 5, Figure 5 Page 6, discussion Page

6) published in Rosselli et al. (2015). Again it would be very interesting to merge the data obtained in 2020 and 2015 to confirm the presence of such a conserved core microbiome. Could the authors make this analysis with the integrated data.

Finally, in this paper (2020), Rosselli et al. efficiently exploit wind rose graphs integrating wind speed and direction, temperature and humidity (Figures SM9, 10, 11, 12, 13, 14) to explain some of their results. Actually these wind rose graphs are presented for March, April, May, June, July, August, September, October and November. Unfortunately, data are not presented for February, a time period of interest for the experiments reported in Rosselli et al. (2015). Could the authors add these data and comment about the results of 2015.

**Other comments**

It would be interesting to give the total number of cells present in the various samples as it is also a very important indicator describing the air microbiome. Does this number differ depending on the situations (sampling site, season, wind..)?

Fig S1 and S4: the authors present data concerning the amount of PM10, do they have data on PM 2.5? It would be interesting to look at them and see if there is a variation of their concentrations with the seasons, locations, dust events …etc.

The authors refer to the importance of the "daytime height of the planetary boundary layer over Europe " (Page 26, line 543). This is indeed an important factor that can shape the air microbiome. Do you have data on the height of the boundary layer at the sampling sites and during the air mass trajectories? It would be of great interest to add it to this manuscript and take it into account.

Finally, I found some mistakes in the citation of the references:

*Gleick et al (1993, Page 2 line 48) and Shine et al. (2000, Page 3 line 69) are not in the reference list.

*Page 3 line 64 "Polymenakou "(and not "Polimenakou").

*Some references in the list are not cited in the text:

Harland et al, 2008 (page 34, line 748)

Koenig et al, 2010 (page33 line 706)

Kramer et al, 2006 (Page 35, line 768)

Latif et al, 2014 (Page 35, line771)

Poschl, 2006 (page 37, line 808)

Shao et al, 2011 (page 38, line 834)

Shiklomanov et al, 1993 (page 38, line 837)

Wainwright et al, 2003 (page 39, line 867)

---

## Referee Comment (RC2) · Anonymous Referee #3 · 7 Feb 2021

General Remarks: The manuscript describes a study on the parameters affecting airborne microbial community composition, e.g., season, dust intrusion, geographic proximity to the dust source. These are important questions in the study of aerobiology, especially in the Mediterranean basin that is prone to increasing frequency of Saharan dust intrusions. The study presents a surprising result, according to which the time of the sampling is the most significant factor affecting the airborne microbial community composition. Although seasonal differences have been demonstrated in previous

studies, at various locations, I have no knowledge of any that have resulted in such overwhelming differences between two sampling campaigns at the same location, under similar atmospheric conditions. This does not come to doubt the validity of the results; however, extra-caution should be taken to ensure that no confounding variables are responsible for this result. A possible cause for this result might stem from batch effects, e.g., DNA extraction, amplification and sequencing conducted by two different people, on two different occasions might be sufficient in producing such differences. Therefore, the authors are urged to specify whether actions were taken to prevent any batch effects.

Other general suggestions: 1. The term "seasonality" can be used if a cyclic change over seasons is shown, the difference between May and September of a single year is better referred to as "temporal".

2. Only a single sample per location per month represents the ambient conditions, therefore it is hard to compare dusty to clear conditions. In the absence of several control samples per site, per month, one cannot appreciate the natural variation of the airborne community. With the current study design, the samples representing the same month or the same site cannot provide information on dusty vs. clear days. Possible comparisons can only be made between sites and between sampling periods (September / May). Clear to dusty conditions can only be compared across the entire dataset. However, the great variance observed between May and September probably masks the role of dust in changing the atmospheric bacterial community.

Specific remarks:

L. 32: "concerted succession..." – The use of the term "succession" implies bacterial growth and selection. Please rephrase throughout the manuscript.

L. 54: particle size can well exceed 10 um. Dust storms often carry larger particles (Ryder et al., 2018).

L. 61: Please provide a specific website address, the home page of WHO is insufficient.

L. 73-76: Should rephrase: according to the cited paper these genes are not specific to atmospheric bacteria, it is suggested that their presence might enable bacterial survival in the atmosphere.

L. 88: Change "until" to: "up to", or: "reaching".

L. 149-151: this is not so clear. What is the filtering step? What do the two filters represent? A single sampling event? Two consecutive sampling days?

The methods section should clearly indicate how many samples were collected, what was the duration of each sampling event, their dates, etc. It is advised to add a table that sums all the sampling data. DNA extraction and sequencing: The choice of relatively large amplicons (∼1400 bp) along with 93 cycles in a paired-end sequencing is surprising. Although there are various platforms that allow pre-processing of non-overlapping sequences, this might introduce more errors to the alignment step and to the taxonomic classification. Could the very high number of counts per sample (min. 1109571) arise from some sequencing pre-processing error? Were all samples sequenced on a single lane? Were they separated to different lanes according to their dates? Were the amplification and/or sequencing conducted in batches?

L. 182: "OUT table" should be "OTU table"

Figure 3: Please increase the font and provide clear titles and a color legend within the figure, and not as part of the caption. In general, pie-charts tend to be less clear than bar-charts, since the human eye estimates height differences better than area differences. It is possible to create a dendrogram including all samples, better emphasizing the clustering of samples according to the time of sampling. Also, does each pie-chart represent a single sample or a group of samples? It is not clear how many samples were used for this study.

Section 3.2: The authors describe in detail the differences observed between the samples. These samples represent an array of conditions: dust events vs. clear days; September vs. May; Cagliari vs. Sassari. A clearer representation might be achieved if this section was divided and each comparison was described separately.

L. 315-321: The authors assume that taxa that will show the least variability between two samples of the same dust event, belong to a local core microbiome. This assumption should be better explained, and answer the following questions: 1. Since dust events introduce new taxa, they are expected to "dilute" the core microbiome, resulting in a variation between dusty/clear conditions. Why not look for the bacteria that decrease in abundance when dust events occur? 2. Comparing two samples of 12 hrs each of a single dust event seems arbitrary. If PM10 mean concentrations throughout the examined 24 hrs remained stable, more dust-related taxa would show low variance, and be considered as "core microbiome". How do the authors assure that this is not the case here?

There is some confusion regarding which taxa were considered as significant for this analysis – L. 316 states that "The latter were considered to represent the common core...", referring to the taxa that showed lower variation. Following, on L. 320, it is stated that: "Only taxa which displayed a mean variation higher than 1/2 of the corresponding standard deviation were considered." In the following tables' captions it is stated that the numbers represent taxa that exceeded the threshold. Please clarify what is the aim of this analysis – to find a core microbiome or to find the "immigrant" bacterial community.

Tables 1 and 2: On table 1 the rows' order represents the two locations, alternately; on Table 2 the rows' order represents first Sassari and then Cagliari. Uniformity between tables is advised.

Table 3: This is the first clear indication of the number of samples taken at each site, under the selected conditions.

Figure 4: When the data is expected to vary along several parameters, it is advised to

examine other PC axes, e.g., PC3 and PC4. This way, the effect of the most influencing variable (in this case – the time of the sampling) is less expressed, and gives way to see other parameters, possibly. Instead of showing the same PCA triplicated, the authors can attempt to make a more condense view of PC1 vs. PC2, using shapes and colors etc., and add figures displaying other PC axes, if they indeed show some correspondence to the other examined variables.

Figure 5: There's some redundancy in figures and tables: there is no new information arising from the PCoA in Figure 5, that wasn't provided by the PCA in Figure 4. Table 3 provides in detail the diversity indices of the samples, and Figure 5 A and B display it as a boxplot. There's no need for both, the table can be moved to the supplementary information.

L. 416: Using a Bonferroni correction for multiple comparisons is unnecessarily stringent. It resulted in only 6 taxa that were significantly differentially abundant when comparing the two sampling periods. A result of only six significant genera is probably of a low ecological relevance. It is more advised to apply Benjamini-Hochberg correction to achieve better statistical power, and a greater collection of significant taxa. It is often the case with very low p-values, that they represent rare taxa that are found in very few samples. I advise the authors to inspect the six significant taxa, and make sure this is not the case here. Moreover, other statistical tools, better fitting microbiome data analysis, are available for differential abundance tests, e.g., ANCOM, ANCOM-BC, MaAsLin2, etc. These methods should be preferred over ANOVA tests for compositional datasets.

Table 5: This table is somewhat overcrowded and demands an intense reading to draw conclusions from. It presents pair-wise distances between individual samples, and not between groups of samples (May vs. September, CA vs. SS, etc.), which provides no statistical significance to the observations arising from it. It also seems somewhat redundant, there is no added value in this table over the dendrogram, PCA and PCoA already presented.

L. 473: the declared goal of the study is indeed significant to the understanding the processes affecting the airborne microbial community in the Mediterranean; however, the design of the study suffers from too few samples. There are too few controls, and only two sampling campaigns, representing two seasons. Seasonality cannot be determined without repetitive sampling during the same season over several years. The presented differences can be referred to as temporal.

L. 484: This remark is very true.

L. 488: This finding is very different from what was suggested by others before, e.g., Gat et al. (2017); Lang-Yona et al. (2020); Bowers et al. (2013); Caliz et al. (2018). Seasonal variations were shown in previous studies, yet they were not as extreme as shown in this study. Sampling location and aerosol back-trajectories were usually more significant in determining the airborne community composition. Did the authors make sure that no batch effects or other confounding variables stand behind the results presented here?

L. 527-532: The referral to PM10 concentrations is significant, especially when considering the changes in community composition during dust events. According to Figure S1, the dust events on May and September have tripled the atmospheric PM10 concentrations, compared with clear days. This significant change in PM10 is expected to result in a significant change in the airborne bacterial community (see Mazar et al., 2016). Yet, this change seems minor according to Fig. 3, 4 and 5. How do the authors explain this result?

L. 551-554: As stated before, sometimes the lowest p-values are given by the rarest taxa. Please make sure this is not the case for these taxa.

L. 576-593: Due to the low number of control samples, it is impossible to draw conclusions on dusty vs. clean conditions of the same season or the same location.

---

## Referee Comment (RC3) · Youssef Yanni (Referee) · 11 Feb 2021

This is a track changes version of the text containing some notes and corrections that can improve the quality of the abstract and can through light of the revision needed for the entire parts of the context. I hope that they can assist production of a revised version suitable for publication. The revised text is also attached to the email for just in case the following text is not fully informative!

[Figure]

Abstract. An NGS-based taxonomic analysis was carried out on airborne bacteria sampled at ground level in two periods (May and September, XXXX) and two opposite localities on the North-South axis of the Sardinia Island (Between Latitudes XXXXXX to XXXXXX). Locatinged around thein a central position of the Mediterranean basin, Sardinia constitutes a suitable outpost to reveal possible immigration of bacterial taxa during transcontinental particle discharge between Africa and Europe. TWith the aim was to of verifying relative effects of dust outbreaks, sampling period and sampling site, on the immigratingairborne bacterial community compositions, we compared the bacterial loads of air collected during dust-carrying meteorological events to that coming from wind regimes not associated to long-distance particle lifting. Results indicated that: (a) a higher microbial diversity (genera or sp.) (118 orders vsvs. 65) and increased community evenness were observed in the campaign carried out in September in comparison to the thatone ofin May, irrespective of the place of collection and of the presence or absence of dust outbreaks,. ( b) dDuring the period of standard wind regimes without transcontinental synchronous outbreaks a synchronous, concerted succession of bacterial communities across distant locations of the same island, accompanied as mentioned by a parallel rise in bacterial diversity and community evenness appears to have occurred,. (c) changes in wind provenance could transiently changed community composition in the sampling locality placed on of the coast facing the incoming wind, but not in the one located at the opposite side of the island,; for this reason thus,d) the community changes brought from dust outbreaks of African origin wereare observed only in the sampling station exposed to south; (ed) the same winds outbreak, once proceeding over land appear to uplift bacteria belonging to a common core already present over the region, which diluted or replaced those that were associated with the air coming from the sea or conveyed by the dust particulate, explaining the two prior points, f. (e) the hierarchy of the variables tested in determining bacterial assemblages composition results are: sampling period » ongoing meteorological events > sampling location within the island.

---

## Referee Comment (RC4) · Anonymous Referee #4 · 17 Feb 2021

General Remarks: The authors present the results of an interesting experiment, sampling airborne microbial communities at two locations, Sassari and Cagliari, located at the opposite ends of the Mediterranean island of Sardinia. The first facing Europe in the north, the second facing north Africa, in the south. The study of airborne microbial communities it is of great interested nowadays as the roil of air dispersal in determining the biogeography of micro-organisms is slowly being discovered. The authors collected samples in two different seasons, at the two different locations and considering different dust events/wind direction shifts and compared the results of these variables on the sampled airborne microbial community through the use of amplicon sequencing. The idea of using these two locations is very interesting, since it can clearly capture the airmasses coming from two different continents with less interference. The results show interesting patterns, defining the season as the most important variable in defining alpha and beta diversity differences among the samples. Moreover, the location, although less clearly, seem also to influence combined with the timing of the samples used as control (before and after the dust event). In general, I think the experiment and the results are interesting but I would advise to authors to review certain sections before publication.

Specific comments: 1)The manuscript presents itself with a title that seems way more inclusive, as "Determining the hierarchical order by which the variables of sampling season, dust outbreaks occurrence, and sampling location, can shape the airborne bacterial communities in the Mediterranean basin" points to a much larger study, where the whole Mediterranean basin (and not only the island of Sardinia) is considered, and more observations and samples are available. In this case the experiment is quite small, only having two seasons, and one year time spawn, with only two locations, and it should be rather presented as a case study, since one cannot draw any certain conclusions from these observations, but only hypotheses. This is not meant in diminishing the value of the study since sampling airborne microbial communities presents technical difficulties that make the design of bigger studies quite challenging. Nevertheless, a better title should be considered.

2) The experimental design is not really clear until the results section. A table or a schematic of the design should be included at the beginning of the material and methods section.

3) The result section seems very messy and hard to read. First the authors introduce the readers to the microbial composition: From line 259 to 309 the authors make a great efforts in describing the different microbial taxa that compose the samples. Nev-

ertheless, it is hard to find the logic the authors followed to write this section. Such analysis should start from a clear graph showing the different phyla composing the different samples. The graph provided (figure 3) is extremely hard to read. From there you could enter specific phyla that you want to further comment. Then the authors present a method to identify the taxa that form a common core in the samples. The method they use, in my opinion, is not clearly explained: Line 319 to 321: "the criterion was to set a cutoff value with respect to the percent of variation occurring between the first 12 h of the collection time and the second half of it. Only the taxa which displayed a mean variation higher than $\frac{1}{2}$ of the corresponding standard deviation were considered" So why variation WITHIN the dust event should provide this information? Shouldn't it be better to just analyze the OTUs found in all samples? Not clear why this method was used. Generally, the results rection is hard to follow and confusing and needs a deep restructuring. Further comments below in the technical corrections.

4)The discussion part is well organized. It very well explains the hypotheses of our authors to explain the results observed in relationship to the geographical and meteorological conditions. The authors could nevertheless improve the ecological value of the discussion, better linking previous experiments and observations done with microbial dust dispersal. Moreover, the discussion is mainly based on the measurements of alpha and beta diversity but does not include much of taxonomical data. So, the authors fail to discuss the results explained in the results section (259 to 309, tab 4) Only few lines are dedicated to this purpose (554-556).

Technical corrections:

Line 52: Dispersal not Dispersion. Dispersal is the act itself; dispersion is the result of dispersal.

Line 63: Here it seems the topic changes suddenly. Entering deep into the microbiology. I would put the paragraph ending here and not in line 60

Line 182: OTU

Figure 3: Replace it with a stacked barplot

Line 310 to 314. This might go up in the methods. Once again, a clear section of the methods describing the methodology is needed.

Line 315: OTUs or taxa? Along all the manuscript I had the feeling these words were not used consistently, please check.

Tab1: It is unclear to me what you are reporting here, please explain better.

Tab2: Are you reporting the number of orders? If it is just the count, that is a measure of richness, and not diversity. Also, are these results adding important information compared to the bray Curtis similarity table? It seems quite redundant and not helpful for the discussion, since the number of orders is not really helpful for any results. For quantifying community variations, Bray-Curtis is the right instrument, as used later.

Tab3: Simpson and Shannon are both diversity indexes, and the use of two of them does not add much information to the discussion. Why not using richness instead of Simpson?

Tab4: should be connected to the results at line 259 to 309. But it is not.

Figure 5A, the PCoA does not add any info to the study. Moreover, the dissimilarity matrix used for the PCoA should be reported.

Tab5: This is hard to read. I suggest a heatmap style table.

565: How was it verified? You would need additional controls after and before the dust events.

---

## Author Comment (AC1) · 12 Mar 2021

Anonymous Referee #2 Review bg-2020-324 : . General comment Rosselli et al. (2020) present a paper entitled ÂńDetermining the hierarchical order by which the variables of sampling season, dust 2 outbreaks occurrence, and sampling location, can shape the airborne bacterial communities in the Mediterranean basinÂż. The presented results are of great interest as there is an urgent need of data about bioaerosol's

biodiversity and transport together with the understanding of the parameters shaping the air microbiome The case studies presented in this paper are very well chosen as they consider different situations, all taken in Sardinia with sampling on two different sites: Sassari (North Sardinia) exposed to European influence, Calgari (South Sardinia) exposed to African influence. Also two time periods have been considered: May and September, as well as dust and clear sky events. NGS-based taxonomic analysis has been carried out in all these samples and compared using various pertinent statistical tools. The results allowed to propose a hierarchy of the variables determining the biodiversity of the collected bioaerosols: sampling season » ongoing meteorological events > sampling site. The sampling period had clearly the major impact. This paper is worth publishing in Biogeosciences when the authors answer some major remarks and possibly make new complementary analyses.

ANSWER: We appreciate the very positive remarks expressed and the praise to the topic interest and overall report's soundness and to add that the paper is suited for a publication in Biogeosciences.

Major concerns My major concern lies on the fact that the authors did not compare in an extensive way the results obtained in this study with those previously published by Rosselli et al (Sci Rep. 2015). In my opinion they should have included these first results obtained at the same location sites (Calagari and Sassari), under dust and clear sky conditions but at a different season (February). The same data sets recorded by the same authors are available and could be compared directly with the present one using the same statistical tools. This comparison would bring strong arguments to generalize the hierarchy of the variables influencing the structure of the microbial communities and make the conclusions clearer.

ANSWER: We are grateful for recognizing the relevance of our 2015 published Scientific Reports article; the comment is acknowledged and will be fully addressed below. As regards the first advice (to include those data and results in the present manuscript and join the datasets to include the published ones as well as a whole analysis) the

limiting guideline is the correct policy of Biogeosciences, which prescribes as require-ments: "The work submitted for publication has not been published before, except in the form of abstracts, preprints, published lectures, theses, discussion papers, or sim-ilar formats that have not undergone full journal peer review, and it is not under con-sideration for peer-reviewed publication elsewhere" Another caveat, from the technical side, in relation to comparing datasets from different experiments is that, even when all methods were the same, our two sets of samples were not sequenced in the same flow cell run, which per se brings about a possible batch effect that could introduce some variation bias. The same problem applies in fact to any attempt to compare data with those existing in literature which were obtained by either one's own research or studies from other authors. Nonetheless, we had quoted the key aspects of our prior paper and commented the consistent points as follows: "In our prior work (Rosselli et al., 2015) we had studied community composition in the same Sardinian stations in a short period of winter (in late February) during and after a single dust-carrying event. In that study the main feature evidenced was the existence of a conserved core microbiome, encom-passing 86-95 % of the taxa, to which the incoming dust would cause some detectable diversity variation but on a rather limited proportional scale. Such minor effect of the dust-lifting storms observed in winter is in fact confirmed in the present work in which the time of the year factor appears as the variable of major order in shaping community structure and richness." And we have now added further discussion on that, as we will outline in the answers to the next queries.

The discussion presented in the present paper has to be consistent with the one pub-lished earlier to be really acceptable and sound. For instance, the present paper (Rosselli et al. 2020, Table 3, Page 17) clearly shows that dust events have little influ-ence of the biodiversity indexes (Simpson 1-D, Shannon H), this is quite contradictory with what is claimed in Rosselli et al. 2015 (Table 2, Page 4).

ANSWER: In the present paper, as regards the diversity or evenness indexes, the only differences which have been remarked and claimed as statistically significant are the

ones across the two seasons (Formerly Fig.5B and Fig 5C and currently Fig S9B and S9C, Supplementary); having respectively p< 0.00584 and p<0.026. As regards the same indexes, Table 3 of the present manuscript lists the values of each sampling but not the statistical significance comparison among each. In that case the dust vs. control condition did not result significantly differing. If this outcome has to be compared with the prior report, it needs to be noticed that, in that case the data shown in Table 2 of that publication, were instead not subjected to statistical significance analysis and the apparent difference of higher diversity in relation to the first and single dust outbreak analyzed at that time, in one of the localities (Cagliari) was even showing opposite results when using Shannon or Simpson indexes (increasing in one case and decreasing with the other). Therefore: 1) The present data cannot be considered in disagreement with the prior report as in that case the statistical analysis on those ecological indexes values fluctuations had not been performed; 2) The present data are instead backed up by appropriate analyses, repeated for two independent dust outbreaks, and consequently deemed as much more robust.

Also this dust influence is illustrated in PCA and Cluster dendrograms presented Figure 4 (Page 5) and 5 (Page 6) of the results published in 2015 (Rosselli et al.), again this is contradictory with figures 4, 5 and 6 (Pages 18, 19 and 21 respectively) of the present manuscript. Could the authors comment on these results and possibly merge the data of the two papers with new statistical analyses integrating all the data. It is quite important to clarify the influence of not of dust events on the microbiome compositions.

ANSWER: In the 2015 paper the PCA (which was the only multivariate analysis in that occasion) and the dendrogram, had been plotted by ordering data according to the on Euclidean distance, with average linkage method based on the identified genera. Notice also that in that old paper the percentage of variation explained by each axis was not shown (and it was actually rather low). In the current manuscript instead, only the cluster analysis (Fig.3) made use of that method. While for multivariate analyses

(object of this query): a) the PCA was obtained using the Bray-Curtis distances which are much more appropriate than the Euclidean one in multivariate ordinations when dealing with data in which some taxa feature zero presence in some samplings. b) the percent of explained variability of the two axes shown is reported, amounting to 35%+14% for the PCA of Fig. 4, and to 54% +14% for the PCoA of Fig. 5. (c) raw data had been transformed by the combination of total sum scaling with square root transformation which is the renowned 'Hellinger transformation'. Such transformation has been praised as a preferable choice in ecological community comparisons, (Legendre and Legendre, 1998, Numerical ecology, 2nd English edn. Elsevier, Amsterdam) as it offers the best trade-off between linearity and resolution in comparison to chi-square metrics and other approaches. It is also recognized as more balanced for the weight given to rare species. For this reason it is also the first recommended choice of data transformation in the Calypso webtool suite (Zakrzewski, et al., Bioinformatics 33, 782–783,2017), which has to date used and cited by 292 articles indexed in Web of Science. As regards figs 5 and 6 of the present manuscript, those analyses (Principal Coordinate Analysis and Discriminant Analysis of the principal components) had not been run in the Rosselli et al., 2015 published data and the comparison for their outcome is not feasible. In essence we view the present data as much more reliable than those stemmed from the previous report and at the same time not showing conflicts of compliance with them due to the more limited methods used in the prior paper and the absence of some statistical significance tests thereby used.

Another point concerns the evidence of a "major conserved core microbiome" that could be considered as a "global Sardinian air microbiome" ( Figure 3 Page 5, Figure 5 Page 6, discussion Page 6) published in Rosselli et al. (2015). Again it would be very interesting to merge the data obtained in 2020 and 2015 to confirm the presence of such a conserved core microbiome. Could the authors make this analysis with the integrated data.

ANSWER: while, as mentioned, we could not in the first place use the published campaign's analyses due to the journal's policy, there are other important considerations that can be made to reconcile any apparent doubt between these two reports and we are grateful to the reviewer for invoking this clarification which leads to this addition to the manuscript's text in the discussion section, commenting on the full consistency between the two reports: "In that prior analysis of ours the existence of a common core microbiome of the investigated area was one of the suggested evidences. That concept was stemming from the analysis run in February, therefore towards the end of a winter period throughout which Europe experiences its minima in terms of temperature-driven air turbulence events and as consequence receives more limited influxes of air travelling from seas to land. In the present analysis, we observe that, in spite of the major changes brought about by the temporal factor, the two sampling stations at opposite corners of the 270 km-long island shared the closest level of community composition when they were compared at the same time (see CA Ctrl vs. SS Ctrl in Fig.3 Fig.4, Fig. S9, and Table 5). Moreover, this similarity was maintained in May even though the two control stations were compared after the dust outbreak. Foremost, those two distant sites achieved the maximum of community overlap in September, when controls were compared right before the next outbreak, after a 109 days-long period without such events. During that time the air microbiome of the whole area appears to have changed profoundly, but in a concerted fashion, leading to a high uniformity across the island. These data confirm the view of the prevalence of a core microbiome, as emerged in our 2015 report and add the evidence that such extended core community undergoes also a temporally related concerted turnover. Whether or not this could be also a seasonal (regularly recurrent) phenomenon, will have to be demonstrated by further research on this subject."

Finally, in this paper (2020), Rosselli et al. efficiently exploit wind rose graphs integrating wind speed and direction, temperature and humidity (Figures SM9, 10, 11, 12, 13, 14) to explain some of their results. Actually these wind rose graphs are presented for March, April, May, June, July, August, September, October and November. Unfortunately, data are not presented for February, a time period of interest for the experiments reported in Rosselli et al. (2015). Could the authors add these data and comment about the results of 2015.

ANSWER: The plot for the quadrant of February, also in line with the above new paragraph, is actually rather similar to that of March as during the December through March winter months southern Europe as well as general Europe air circulations and corresponding climate tend to be relatively stationary. Therefore, we have checked, but we find that the February 2014 image would not actually add particular interpretive clues to the big picture. Actually, we find that a much more predictive comparison element is the day air mass backward trajectories calculated by the NOAA HYSPLIT model shown in Fig. 2 of the present manuscript and in Fig. 1 of the prior paper.

Other comments It would be interesting to give the total number of cells present in the various samples as it is also a very important indicator describing the air microbiome. Does this number differ depending on the situations (sampling site, season, wind..)?

ANSWER: The number of microbial propagules, cells, clusters of them, spores or other quantitative aspects of this type is not possible to assess as, even under microscopy-aided operations as it would imply counting objects which are not visually distinguishable from their background of particulate dust and any non-biotic debris. Even under epifluorescence by vital stains, the problem lies in the stratified layers of target and non-target objects that physically mask the visibility of the former and do not allow the unobstructed two-dimensional counting that would be required for such task.

Fig S1 and S4: the authors present data concerning the amount of PM10, do they have data on PM 2.5? It would be interesting to look at them and see if there is a variation of their concentrations with the seasons, locations, dust events . . .etc.

ANSWER: PM10 is in fact including also PM 2.5 as the class is measured as <10 $\mu$m diameter size. The choice of the former is that the latter would not encompass the majority of airborne microbial cells and PM10 is therefore deemed as more informative for such information.

The authors refer to the importance of the "daytime height of the planetary boundary layer over Europe " (Page 26, line 543). This is indeed an important factor that can shape the air microbiome. Do you have data on the height of the boundary layer at the sampling sites and during the air mass trajectories? It would be of great interest to add it to this manuscript and take it into account.

ANSWER: That value is not a uniform measure that could be easily taken nor officially found for a defined given area as it fluctuates continuously in relation to the uprise of the sun towards the zenith and it is affected at local microscale by factors as scattered clouds resulting in ground shadows that are receiving less radiation and so on. Moreover, in the context of the interpretation given in the text we are not referring to the condition occurring just during the days of air sampling but throughout the whole mid-to-late summer period which is however always bound to result in averagely higher values of hot air convective turbulence in comparison to the colder spring times. The comment was essentially about the observation that late summer in Southern Europe constitutively carries every year higher loads of uplifted microbial cells.

Finally, I found some mistakes in the citation of the references:

*Gleick et al (1993, Page 2 line 48) and Shine et al. (2000, Page 3 line 69) are not in the reference list. *Page 3 line 64 "Polymenakou "(and not "Polimenakou"). *Some references in the list are not cited in the text: Harland et al, 2008 (page 34, line 748) Koenig et al, 2010 (page33 line 706) Kramer et al, 2006 (Page 35, line 768) Latif et al, 2014 (Page 35, line771) Poschl, 2006 (page 37, line 808) Shao et al, 2011 (page 38, line 834) Shiklomanov et al, 1993 (page 38, line 837) Wainwright et al, 2003 (page 39, line 867)

ANSWER: We thank you for the careful attention. The two missing references have been added and the errors fixed. Koenig was not a reference on its own but a carriage return line of the Caporaso et al.

---

## Author Comment (AC2) · 12 Mar 2021

General Remarks: The manuscript describes a study on the parameters affecting airborne microbial community composition, e.g., season, dust intrusion, geographic proximity to the dust source. These are important questions in the study of aerobiology, especially in the Mediterranean basin that is prone to increasing frequency of Saharan

dust intrusions. The study presents a surprising result, according to which the time of the sampling is the most significant factor affecting the airborne microbial community composition. Although seasonal differences have been demonstrated in previous studies, at various locations, I have no knowledge of any that have resulted in such overwhelming differences between two sampling campaigns at the same location, under similar atmospheric conditions. This does not come to doubt the validity of the results; however, extra-caution should be taken to ensure that no confounding variables are responsible for this result. A possible cause for this result might stem from batch effects, e.g., DNA extraction, amplification and sequencing conducted by two different people, on two different occasions might be sufficient in producing such differences. Therefore, the authors are urged to specify whether actions were taken to prevent any batch effects.

ANSWER: The possible batch effect issue does not appear to apply in this case as the processing has been done by the same single operator in all sampling times. A chart with the data details showing the processing uniformity of the throughput is provided in response to a further comment below and is available as Supplementary Table S2.

Other general suggestions: 1. The term "seasonality" can be used if a cyclic change over seasons is shown, the difference between May and September of a single year is better referred to as "temporal".

ANSWER: We find the comment very appropriate; as a matter of fact we have changed also the title of this manuscript by substituiting the word ‘season’ with ‘period’ (The new title is: Determining the hierarchical order by which the variables of sampling period, dust outbreaks occurrence, and sampling location, can shape the airborne bacterial communities in the Mediterranean basin). Moreover the term has been corrected throughout the manuscript in whichever occasion it had been used to refer to this single-year campaign and not to truly recurring phenomena. The use of the words season or seasonal in the paper is therefore limited to the descriptive context, while in any instance in which we infer/suggest interpret something from the observed data, we

are now using the terms time, temporal or period.

2. Only a single sample per location per month represents the ambient conditions, therefore it is hard to compare dusty to clear conditions. In the absence of several control samples per site, per month, one cannot appreciate the natural variation of the airborne community. With the current study design, the samples representing the same month or the same site cannot provide information on dusty vs. clear days. Possible comparisons can only be made between sites and between sampling periods (September / May). Clear to dusty conditions can only be compared across the entire dataset. However, the great variance observed between May and September probably masks the role of dust in changing the atmospheric bacterial community.

ANSWER: One aspect that needs to be considered here is that an atmospheric sampling is not to be regarded with the same conceptual metrics that would apply if one were to study liquid environments as e.g. a seashore or solid ones, as a farm plot, for which cases 'one' sample could correspond to the 50 ml filling a falcon tube dipped in water, or a gram of soil scooped from the ground. In skypost air filtering the operation is carried out continuously for days and one sample, in our case, accumulates the content of 56160 liters of continuously changing atmosphere, which takes into account the variations that occur during all those hours, inclusive of the day/night shifts. One sample is therefore not a 'point' but a built-in averaged replication protocol for the chosen window of events. Moreover, since it is already known from literature that, as microbial community composition is concerned, even in the absence of dust outbreaks, the ambient state of the atmosphere is not stable either, our goal was not to compare an hypothetical status quo with an altered one. The meaning of the 'control' here was to catch 1) the first possible timeframe after the stopping of a northbound dusty wind outbreak (it occurred in May) or 2) the latest possible timeframe of a situation before the onset of a dust-carrying change of wind regime (it occurred in September). Thence, in the latter event the control is not intended as a situation of calm that could represent a period of unknown length, but rather the time-zero sample of the dust event itself.

While for the former case in May, the control is symmetrically designed as the quiet after the storm.

Specific remarks:

L. 32: "concerted succession..." – The use of the term "succession" implies bacterial growth and selection. Please rephrase throughout the manuscript.

ANSWER: The term was corrected.

L. 54: particle size can well exceed 10 um. Dust storms often carry larger particles (Ryder et al., 2018).

ANSWER: The size class limit has been corrected and the reference added.

L. 61: Please provide a specific website address, the home page of WHO is insufficient.

ANSWER: The correct references (instead of a website) have been placed (Prospero et al. 2002, Schepansky et al. 2007) L. 73-76: Should rephrase: according to the cited paper these genes are not specific to atmospheric bacteria, it is suggested that their presence might enable bacterial survival in the atmosphere.

ANSWER: Correction made

L. 88: Change "until" to: "up to", or: "reaching".

ANSWER: Correction made (up to).

L. 149-151: this is not so clear. What is the filtering step? What do the two filters represent? A single sampling event? Two consecutive sampling days?

ANSWER: The sentence was still referring to the sampling (filtering air for 24 h). The two filters are parallel replicates treated independently up to the DNA extraction. The two consecutive sampling days was incorrectly written to mean the two periods of 12 h each in which the sampling was divided during the dust outbreaks. The text was edited to clarify the procedure as follows: “The experimental design involved: 2 sampling sites

at the opposite corners of the Sardinia island (Sassari vs. Cagliari), 2 sampling periods (May vs. September) 2 meteorological conditions (absence vs. presence of a dust outbreak). In each of these situations, two replicate samples were taken and processed independently throughout the DNA extraction step to be pooled before sequencing. Samples were collected on Teflon filters (Sartorius Stedim Biotech) by using a Skypost Tecora apparatus (compliant to the European legislation 96/62/gmeCE) processing 39 liters of air per minute. To constitute 'a sample' a continuous 24h-long air intake through the filters was performed. In the case of the dust outbreaks the 24h sampling was further divided in two periods, by considering independently the first 12 hours and the second 12 hours. The number of resulting samples was therefore 12; namely the module of three conditions: (a) Control; (b) Dust h 1-12, (c) Dust h 12-24; multiplied by 2 sampling periods and by the 2 sampling places, resulting in 3 x 2 x 2= 12 samples. As technical note, the scope and meaning of 'controls' here was that of samples that could allow to individuate the shift between one condition and its adjacent one. In our cases, catching the sudden change of wind regime by sampling immediately before or after a dust outbreak. Therefore, the controls were thus defined as single 24h time points flanking the key dust events."

The methods section should clearly indicate how many samples were collected, what was the duration of each sampling event, their dates, etc. It is advised to add a table that sums all the sampling data. DNA extraction and sequencing:

ANSWER: besides the above editing of the text we have included in the Supplementary material as indicated, a table showing the distribution of the reads output throughout the sampling which clarifies the homogeneity of the protocol outcome and addresses the queries on the possible batch variability ruling out such possibility. (Table S2. Details on the sequencing output quality and evenness of distribution across samples)

The choice of relatively large amplicons (_1400 bp) along with 93 cycles in a paired-end sequencing is surprising. Although there are various platforms that allow pre-processing of nonoverlapping sequences, this might introduce more errors to the

alignment step and to the taxonomic classification. Could the very high number of counts per sample (min. 1109571) arise from some sequencing pre-processing error? Were all samples sequenced on a single lane? Were they separated to different lanes according to their dates? Were the amplification and/or sequencing conducted in batches?

ANSWER: The sequencing strategy chosen at the time was using the Nextera XT DNA protocol via a whole amplification of the 16S rRNA and a shotgun sequencing with 93bp x 2 paired-end reads. All the DNA samples have been therefore sequenced in the same flow lane to avoid biases due to different sequencing batches. The high number of reads obtained from each sample is due to the high efficiency protocol for very low amounts of DNA.

L. 182: "OUT table" should be "OTU table"

ANSWER: Correction made

Figure 3: Please increase the font and provide clear titles and a color legend within the figure, and not as part of the caption. In general, pie-charts tend to be less clear than bar-charts, since the human eye estimates height differences better than area differences. It is possible to create a dendrogram including all samples, better emphasizing the clustering of samples according to the time of sampling. Also, does each pie-chart represent a single sample or a group of samples? It is not clear how many samples were used for this study.

ANSWER: Fig. 3 was restructured graphically by enlarging the charts, the fonts, the scale and by adding the color coded taxonomy in the figure itself. As now clarified by the above answers, in the figure each chart represents a single 24h sample (for the dust event samples, the results of the first and and second 12h periods and have been merged in a single pie to compare each case with the same filtering duration). The information has been added in the text and legend.

Section 3.2: The authors describe in detail the differences observed between the samples. These samples represent an array of conditions: dust events vs. clear days; September vs. May; Cagliari vs. Sassari. A clearer representation might be achieved if this section was divided and each comparison was described separately.

ANSWER: Our description was following the microbiology clades as leading topics and for each main phylum or class we comment the differences in their occurrence in relation to the three variables of time, meteorology and geography, by the order which explains the partitioning of each phylum. We have checked the rearrangement of the section as suggested, but in that case, there would be a six-fold multiplication of the description of each taxonomical group (e.g. for Actinobacteria we would need to describe their statuses starting over each time in the six different paragraphs for May, September, Cagliari, Sassari, Dust and Control) and in some cases no relevant facts apply for many phyla. The result appeared to convey a more dispersed view when compared to the presentation of patterns by-microbiology in which we underlined only the variables involved in main differences and we could also group the description of taxa that showed common behavior for different variables. We also find that having redrawn and anticipated Fig.3 and repositioned the former Table 4 (Now table1, allows to follow the text of section 3.2 in a much smoother way.

L. 315-321: The authors assume that taxa that will show the least variability between two samples of the same dust event, belong to a local core microbiome. This assumption should be better explained, and answer the following questions: 1. Since dust events introduce new taxa, they are expected to "dilute" the core microbiome, resulting in a variation between dusty/clear conditions. Why not look for the bacteria that decrease in abundance when dust events occur?

ANSWER: The method indeed lacked clarity and an important premise. Addressing first the comment on why not look at bacteria that decrease in abundance being diluted (passively) by mass immigration of others. The issue has to do with the distinction between actual population dynamics (ecologically-ruled) and mathematical effects of

sampling from a ballot box of objects, all of which compete for the constrained 100% format of results (probability-ruled). The problem arises when comparing communities at different time points (and even more with DNA-based methods with fixed total DNA amount processed), in which results are based on percent values (relative abundance of taxa). In that case the multiplication of any, determines obviously a reduction of the relative abundance of others when those do not grow at equal or higher rate. Therefore one given group could have been increasing, but its share in the sum could appear as if it had instead decreased if a different group has increased faster. This consideration, that applies inevitably for all metagenomics studies, should be kept in mind for all kinds of interpretations about increases and decreases which could be either real or apparent (when driven by a stronger change of a different group). As consequence, comparing different sampling points through time is linked to this inevitable constraint: the compositional nature of the dataset binds all relative frequencies to each other. Therefore, since the sum of them is bound to give always 100% the decrease of a given species could be either apparent (driven by the increase of another), or real (due to its actual negative population dynamics). The problem is that the two causes can not be uncoupled by just comparing their frequencies at the two sampling times. Moreover, even an actual increase of a given species could be masked by the parallel increase of another at a faster pace (or by its net immigration into the scene). This is the reason by which we consider with caution the possibility of looking at decreasing taxa as indicative of their actual biological activities or fate.

As regards the first part of the query, we revised the text to address the comment as follows:

"Besides the comparisons that included all OTUs to put in evidence community variations, in parallel we exploited an additional opportunity to detect possible dust-specific taxa. The rationale was to seek differential enrichment within the dust storm, by dissecting the process, during its progression, splitting its onset from its fully established stage. To this aim we collected separately the filters of the first 12h of the event, and

replaced them with new ones that collected air during the second lapse (hours 12 to 24). Thus the availability of two timeframes, both within the dust event, allowed to verify which OTUs would be incrementally enriched along with the progression of the stormy condition. This allowed to better refine the bacterial deposition dynamics during the outbreaks. From the visual and physical points of view, an increase in the inflow of air particulate was observed for the 12-24 h period, confirming the differential level of deposition occurring in the maturity stage of the meteorological phenomenon. This within-outbreaks set up was essentially aiming at individuating taxa that would display high variation in relation to dust events in comparison to those who would not. The latter were considered to represent the common core of bacteria that were constantly present in samples, irrespective of the changing meteorological events. To apply this distinction, the criterion was to set a cutoff value with respect to the percent of variation occurring between the first 12 h of the collection time and the second half of it. The choice of this threshold was considered critical and, in order to ensure robust conclusions, we deemed necessary to require a considerable consistency of variation. Pointing at this objective, only the taxa which displayed a mean variation higher than $\frac{1}{2}$ of the corresponding standard deviation were taken into account. The resulting level of variation in the two sampling stations is reported in Tab. 2 (Formerly Tab. 1) and the corresponding number of orders is displayed in Tab. 3 (Formerly Tab. 2). The Sassari (North-facing) collection site was the one that in both seasons resulted to feature the highest number of significantly changing taxa. The identities of these are shown in Supplementary Fig.S7 (May event), and Supplementary Fig. S8 (September event). In the graphs, the first 12h lapse is plotted above the baseline and the second (12-24 h) is on the specular position below."

We also added a clearer legend to the Table (former Table 1) as follows:

" Extent of OTUs change across cell harvesting time during the same dust outbreak sampling. The percent variation (either increase or decrease) of a given OTUs abundance between the values found in the community obtained by the first 12h sampling

and the ones resulting from the following 12h lapse was computed. The average, minimum and maximum percent variation between counts are shown. Only taxa displaying a difference in percentages higher than half of their standard deviation were selected for the present comparison."

2. Comparing two samples of 12 hrs each of a single dust event seems arbitrary. If PM10 mean concentrations throughout the examined 24 hrs remained stable, more dust-related taxa would show low variance, and be considered as "core microbiome". How do the authors assure that this is not the case here?

ANSWER: PM10 data by collecting services are made available in delayed fashion, but in our sampling we could decide the splitting not on an arbitrary pre-assumption but as, during preliminary trials in 2013 and in the February 2014 campaign (Published in Ref. Rosselli et al., 2015), we could witness in real time that an increase in the inflow of air particulate was occurring during the second (12-24 h) timeframe.

There is some confusion regarding which taxa were considered as significant for this analysis – L. 316 states that "The latter were considered to represent the common core: : :", referring to the taxa that showed lower variation. Following, on L. 320, it is stated that: "Only taxa which displayed a mean variation higher than 1/2 of the corresponding standard deviation were considered." In the following tables' captions it is stated that the numbers represent taxa that exceeded the threshold. Please clarify what is the aim of this analysis – to find a core microbiome or to find the "immigrant" bacterial community.

ANSWER: As mentioned above in relation to the compositional dataset constraints, the pursuit was to put in evidence differences that would be least affected by the apparent indirect mathematical effect that is common to all these studies. The common core is not the ideal word here as we would better define those as the bulk of taxa that display a behavior which is only indirectly affected by that of the others (i.e. ecologically unaffected and only mathematically affected in their relative frequencies resulting by

the behavior of others). The core microbiome here is to be seen as a complement background within which the immigrants are impinging. These aspects have been further detailed in the text of the section.

Tables 1 and 2: On table 1 the rows' order represents the two locations, alternately; on Table 2 the rows' order represents first Sassari and then Cagliari. Uniformity between tables is advised.

ANSWER: The former Table 1 (now Table 2) was reordered to achieve uniformity

Table 3: This is the first clear indication of the number of samples taken at each site, under the selected conditions.

ANSWER: We agree, having now anticipated the samples outline (3x2x2=12) in the text at above described in the Materials and methods should help to avoid confusion.

Figure 4: When the data is expected to vary along several parameters, it is advised to examine other PC axes, e.g., PC3 and PC4. This way, the effect of the most influencing variable (in this case – the time of the sampling) is less expressed, and gives way to see other parameters, possibly. Instead of showing the same PCA triplicated, the authors can attempt to make a more condense view of PC1 vs. PC2, using shapes and colors etc., and add figures displaying other PC axes, if they indeed show some correspondence to the other examined variables.

ANSWER: We have inspected further axes, however the explained variability added by PC3, PC4 etc. is of marginal increment (PC2 is already as low as 14%) and the representations were not adding clarity. However, the clearest complement to the data shown in Fig.4 is in our opinion Fig.5 (Formerly Fig. 6) which draws on the same PCA data and extracts its information by the discriminant analysis.

Figure 5: There's some redundancy in figures and tables: there is no new information arising from the PCoA in Figure 5, that wasn't provided by the PCA in Figure 4. Table 3 provides in detail the diversity indices of the samples, and Figure 5 A and B display it

as a boxplot. There's no need for both, the table can be moved to the supplementary information.

ANSWER: We agree and Fig. 5 has now been placed in the Supplementary material

L. 416: Using a Bonferroni correction for multiple comparisons is unnecessarily stringent. It resulted in only 6 taxa that were significantly differentially abundant when comparing the two sampling periods. A result of only six significant genera is probably of a low ecological relevance. It is more advised to apply Benjamini-Hochberg correction to achieve better statistical power, and a greater collection of significant taxa. It is often the case with very low p-values, that they represent rare taxa that are found in very few samples. I advise the authors to inspect the six significant taxa, and make sure this is not the case here. Moreover, other statistical tools, better fitting microbiome data analysis, are available for differential abundance tests, e.g., ANCOM, ANCOM-BC, MaAsLin2, etc. These methods should be preferred over ANOVA tests for compositional datasets.

ANSWER: We are grateful for the advice and for the uncommon acknowledgment of having been statistically even too severe. We have inspected the loosening of stringency effect, resulting in a more generalized array of taxa but supported by less robust p values, whose ecology is however less clearly linked to a nexus to the situation and whose presence appears more stochastic. We have also checked that for the six taxa that stood the Bonferroni test, are in all cases either consistently present in all samples or recur with frequencies higher than 1% in them. We therefore would maintain the high stringency output as the one we feel more confident to prudently describe.

Table 5: This table is somewhat overcrowded and demands an intense reading to draw conclusions from. It presents pair-wise distances between individual samples, and not between groups of samples (May vs. September, CA vs. SS, etc.), which provides no statistical significance to the observations arising from it. It also seems somewhat redundant, there is no added value in this table over the dendrogram, PCA and PCoA

already presented.

ANSWER: It is true and we have experimented different ways to reorganize that, including a heatmap-style arrangement with all the entries ending in a single matrix rather than three separate table sections, which however ended up more complex as it needs a tridimensional representation of the same/different season, dust/control and Cagliari/Sassari location. But actually the key message that this color table is meant to convey is just the fact that the low similarity pairwise comparisons (conditionally formatted as red-yellow shaded cells) are almost all observed in the upper table. i.e. the between-seasons comparisons, while the green ones (higher similarities) are all distributed in the two same-period comparisons. We further stressed this aspect to direct the readers' attention to the way to get the bottom line information from this table. The message is reinforced by the other figures as mentioned but this one shows the Bray-Curtis values and allows each pair to be inspected, which is an information that the PCA, PCoA and dendrogram are not providing with such detail. The table serves also a source of the dissimilarity matrix values for the Bray-Curtis based multivariate analyses, that a different reviewer has asked to provide.

L. 473: the declared goal of the study is indeed significant to the understanding the processes affecting the airborne microbial community in the Mediterranean; however, the design of the study suffers from too few samples. There are too few controls, and only two sampling campaigns, representing two seasons. Seasonality cannot be determined without repetitive sampling during the same season over several years. The presented differences can be referred to as temporal.

ANSWER: While as regards the number of control samples we have answered above which was the meaning of them (control as time zero of the dust itself), we fully agree on the fact that we can not interpret as seasonal (recurrent) patterns what we observe in a year and that we should stick to the term temporal to comment these observation and we have modified the terminology throughout the manuscript, including the change of its title.

L. 484: This remark is very true.

ANSWER: We appreciate the positive comment, which was indeed also already addressing the previous (L.473) point.

L. 488: This finding is very different from what was suggested by others before, e.g., Gat et al. (2017); Lang-Yona et al. (2020); Bowers et al. (2013); Caliz et al. (2018). Seasonal variations were shown in previous studies, yet they were not as extreme as shown in this study. Sampling location and aerosol back-trajectories were usually more significant in determining the airborne community composition. Did the authors make sure that no batch effects or other confounding variables stand behind the results presented here?

ANSWER: The cited studies indeed find also variations as acknowledged. In our case, as part of the prior comments'answers, we can not exclude that the single year that we analyzed could represent as particularly variable one and that repeating the same comparisons throughout different years would lead to lower variability as that caught by other authors (some of which in their own single-year analyses). As mentioned above, we can instead rule out the batch effect or confounding issues due to the sampling as it was repeated by the same operator and as the raw sequencing outputs (added supplementary table) do not show evidences of inter-sample variability in terms of technical throughput.

L. 527-532: The referral to PM10 concentrations is significant, especially when considering the changes in community composition during dust events. According to Figure S1, the dust events on May and September have tripled the atmospheric PM10 concentrations, compared with clear days. This significant change in PM10 is expected to result in a significant change in the airborne bacterial community (see Mazar et al., 2016). Yet, this change seems minor according to Fig. 3, 4 and 5. How do the authors explain this result?

ANSWER: Partly by the fact that PM10 is a physical measure of particulate size class

and does not straightly equate with a content of airborne biota. Mostly because the bulk of PM10 over industrialized or inhabited territories includes combustion particulate (which is by its source devoid of microbial cells or intact DNA) and also by the fact that in dust outbreaks a vast majority of the airborne fine material is not loaded with microbes as it comes from airlifted particles from desert zones in which a strong selection is exerted against surface life by unshielded UV radiation exposure, dry conditions and absence or scarcity of primary productivity. Results (as in Mazar et al. 2016) could also be dependent on the distance of the sampling location from the departing site of the airborne material and by the population density of the land crossed before discharge or in the surrounding of the sampling outpost.

L. 551-554: As stated before, sometimes the lowest p-values are given by the rarest taxa. Please make sure this is not the case for these taxa.

ANSWER: The issue has been addressed above and we refer to the prior comment

L. 576-593: Due to the low number of control samples, it is impossible to draw conclusions on dusty vs. clean conditions of the same season or the same location.

ANSWER: The text was rephrased to account for what we relied above to the same issue. Since the airborne community composition does change daily even during periods that do not feature the dust carrying episodes, there is not a stable condition that could be considered as the durable control community. Even the evening and morning opposite breeze regimes that occur daily in coastal locations impart modifications. Therefore taking a series of 'controls' intended as samples in different days before or after a dust storm, would have consisted in just as many different samples. The idea was instead to catch the shift that corresponds to the sudden change of wind regime immediately before or after a dust outbreak. Therefore, as explained. The controls are bound to be single time points that flank the two key events. The concept has been better outlined in the revised manuscript.

---

## Author Comment (AC3) · 12 Mar 2021

Youssef Yanni (Referee) yanni244@yahoo.com This is a track changes version of the text containing some notes and corrections that can improve the quality of the abstract and can through light of the revision needed for the entire parts of the context. I hope that they can assist production of a revised version suitable for publication. The revised text is also attached to the email for just in

case the following text is not fully informative!

Abstract.   An NGS-based taxonomic analysis was carried out on airborne bacteria sampled at ground level in two periods (May and September, XXXX) and two opposite localities on the North-South axis of the Sardinia Island (Between Latitudes XXXXXX to XXXXXX). Locatinged around thein a central position of the Mediterranean basin, Sardinia constitutes a suitable outpost to reveal possible immigration of bacterial taxa during transcontinental particle discharge between Africa and Europe. TWith the aim was to of verifying relative effects of dust outbreaks, sampling period and sampling site, on the immigratingairborne bacterial community compositions, we compared the bacterial loads of air collected during dust-carrying meteorological events to that coming from wind regimes not associated to long-distance particle lifting. Results indicated that: (a) a higher microbial diversity (genera or sp.) (118 orders vsvs. 65) and increased community evenness were observed in the campaign carried out in September in comparison to the thatone ofin May, irrespective of the place of collection and of the presence or absence of dust outbreaks,. ( b) dDuring the period of standard wind regimes without transcontinental synchronous outbreaks a synchronous, concerted succession of bacterial communities across distant locations of the same island, accompanied as mentioned by a parallel rise in bacterial diversity and community evenness appears to have occurred,. (c) changes in wind provenance could transiently changed community composition in the sampling locality placed on of the coast facing the incoming wind, but not in the one located at the opposite side of the island,; for this reason thus,d) the community changes brought from dust outbreaks of African origin wereare observed only in the sampling station exposed to south; (ed) the same winds outbreak, once proceeding over land appear to uplift bacteria belonging to a common core already present over the region, which diluted or replaced those that were associated with the air coming from the sea or conveyed by the dust particulate, explaining the two prior points, f. (e) the hierarchy of the variables tested in determining bacterial assemblages composition results are: sampling period Ảż ongoing meteorological events > sampling location within the island. Interactive comment on Biogeosciences Discuss.,

https://doi.org/10.5194/bg-2020-324, 2020.

ANSWER: We thank the Reviewer for these suggestions marked on the abstract. Apparently the tracked changes ended up mixed with the text, including the one that was meant to be substituted. As consequence, there are a number of typos scattered through the text that the Reviewer has sent, which make many points not easy to interpret. However, if we understood correctly the reviewer's words, these inputs were meant to suggest details to be expanded in the manuscript and we tried to follow these recommendations. Specifically, the geographical coordinates of sampling stations are shown in section 2.1 The year and other temporal details are in section 3.1.

---

## Author Comment (AC4) · 12 Mar 2021

General Remarks: The authors present the results of an interesting experiment, sampling airborne microbial communities at two locations, Sassari and Cagliari, located at the opposite ends of the Mediterranean island of Sardinia. The first facing Europe in the north, the second facing north Africa, in the south. The study of airborne microbial

communities it is of great interested nowadays as the roil of air dispersal in determining the biogeography of micro-organisms is slowly being discovered. The authors collected samples in two different seasons, at the two different locations and considering different dust events/wind direction shifts and compared the results of these variables on the sampled airborne microbial community through the use of amplicon sequencing. The idea of using these two locations is very interesting, since it can clearly capture the airmasses coming from two different continents with less interference. The results show interesting patterns, defining the season as the most important variable in defining alpha and beta diversity differences among the samples. Moreover, the location, although less clearly, seem also to influence combined with the timing of the samples used as control (before and after the dust event). In general, I think the experiment and the results are interesting but I would advise to authors to review certain sections before publication.

ANSWER: We thank the reviewer for the appreciative comments.

Specific comments: 1)The manuscript presents itself with a title that seems way more inclusive, as "Determining the hierarchical order by which the variables of sampling season, dust outbreaks occurrence, and sampling location, can shape the airborne bacterial communities in the Mediterranean basin" points to a much larger study, where the whole Mediterranean basin (and not only the island of Sardinia) is considered, and more observations and samples are available. In this case the experiment is quite small, only having two seasons, and one year time spawn, with only two locations, and it should be rather presented as a case study, since one cannot draw any certain conclusions from these observations, but only hypotheses. This is not meant in diminishing the value of the study since sampling airborne microbial communities presents technical difficulties that make the design of bigger studies quite challenging. Nevertheless, a better title should be considered.

ANSWER: We acknowledge the fact that the landing range is the Sardinia island, which spans 270 km, whose shorelines are 1849 Km-long (1/4/ of the Italian coastline total)

[Figure]

and which we sampled at each end of its main dimension. Nevertheless, the use of the term Mediterranean in the title is referred to the sea (medius-terrae: the sea between lands). This is because, although we used Sardinia as a mid-way catchment sink, the source of the sandy dust that is thereby discharged, encompasses the entire coastal system of the northern African continent, the Saharan atlas as well as the middle east shores and Arabian inland. Therefore, in considering the full transcontinental travel of the passively migrating microbiota, by 'Mediterranean basin' we intended the whole perimeter of origin. As regards the recommended need for more observations e.g. repeating the study in different occasions we can remind that this report is actually the companion follow-up of our prior investigation in which we had carried out the very same approach in the same two places three months earlier, and catching another 'dust vs. calm' shift of events. That one was the first outbreak of that year in the Mediterranean area and yielded the picture that the place featured at the end of the winter (Rosselli et al., 2015: Microbial immigration across the Mediterranean via airborne dust. Scientific Reports 5:16306 DOI: 10.1038/srep16306). The present work is therefore framed in a comprehensive series of analyses, as recommended.

2) The experimental design is not really clear until the results section. A table or a schematic of the design should be included at the beginning of the material and methods section.

ANSWER: We realized this lack of clarity and we found that the design had to be anticipate in the materials and methods in which we added the following description: "The experimental design involved: 2 sampling sites at the opposite corners of the Sardinia island (Sassari vs. Cagliari), 2 sampling periods (May vs. September) 2 meteorological conditions (absence vs. presence of a dust outbreak). In each of these situations, two replicate samples were taken and processed independently throughout the DNA extraction step to be pooled before sequencing. Samples were collected on Teflon filters (Sartorius Stedim Biotech) by using a Skypost Tecora apparatus (compliant to the European legislation 96/62/gmeCE) processing 39 liters of air per minute. To constitute 'a sample' a continuous 24h-long air intake through the filters was performed. In the case of the dust outbreaks the 24h sampling was further divided in two periods, by considering independently the first 12 hours and the second 12 hours. The number of resulting samples was therefore 12; namely the module of three conditions: (a) Control; (b) Dust h 1-12, (c) Dust h 12-24; multiplied by 2 sampling periods and by the 2 sampling places, resulting in 3 x 2 x 2= 12 samples. As technical note, the scope and meaning of 'controls' here was that of samples that could allow to individuate the shift between one condition and its adjacent one. In our cases, catching the sudden change of wind regime by sampling immediately before or after a dust outbreak. Therefore, the controls were thus defined as single 24h time points flanking the key dust events."

3) The result section seems very messy and hard to read. First the authors introduce the readers to the microbial composition: From line 259 to 309 the authors make a great efforts in describing the different microbial taxa that compose the samples. Nevertheless, it is hard to find the logic the authors followed to write this section. Such analysis should start from a clear graph showing the different phyla composing the different samples. The graph provided (figure 3) is extremely hard to read. From there you could enter specific phyla that you want to further comment.

ANSWER: The reviewer is right, the order of presentation of the data and the quality of Fig.3 was in our view most of this problem. We have redrawn Fig. 3 and enlarged, imagery, fonts, added color-coded legend to the taxa in the picture and not in the legend; we increased its overall resolution and we placed the figure before its descriptive comments, which appear now much more sound to follow.

Then the authors present a method to identify the taxa that form a common core in the samples. The method they use, in my opinion, is not clearly explained: Line 319 to 321: "the criterion was to set a cutoff value with respect to the percent of variation occurring between the first 12 h of the collection time and the second half of it. Only the taxa which displayed a mean variation higher than 1/2 of the corresponding standard deviation were considered" So why variation WITHIN the dust event should provide this

information? Shouldn't it be better to just analyze the OTUs found in all samples? Not clear why this method was used. Generally, the results rection is hard to follow and confusing and needs a deep restructuring. Further comments below in the technical corrections.

ANSWER: all OTUs were actually taken into account as well in parallel to put in evidence any possible community variation and this is actually the way most of the results are drawn (Fig.3,4,5, S8, Tab.2,3,4). The difference in this case was to exploit an additional opportunity to detect possible dust-specific taxa. The availability of two incremental timeframes, both within the dust event, allowed to verify which ones would be enriched along with the progression of the stormy condition.

4)The discussion part is well organized. It very well explains the hypotheses of our authors to explain the results observed in relationship to the geographical and meteorological conditions. The authors could nevertheless improve the ecological value of the discussion, better linking previous experiments and observations done with microbial dust dispersal. Moreover, the discussion is mainly based on the measurements of alpha and beta diversity but does not include much of taxonomical data. So, the authors fail to discuss the results explained in the results section (259 to 309, tab 4) Only few lines are dedicated to this purpose (554-556).

ANSWER: The criticism is correct, we integrated the text as follows:

"Commenting on the taxonomical abundance shifts observed between May and September and trying to interpret the rise of some phyla and the drop of others (Fig.3; Tab. 1), a preliminary consideration needs to be recalled. The issue has to do with the distinction between actual population dynamics (ecologically-ruled) and mathematical effects of sampling from a 'ballot box' of objects, all of which compete for the constrained 100% format of results (probability-ruled). This caveat was put forward as early as statistics itself was born as a discipline (Pearson, 1897). There is in this respect a general problem in comparing communities at different time points (and even

more with DNA-based methods with fixed total DNA amount processed), whose results are based on percent values (relative abundance of taxa). In that condition the multiplication of any, determines obviously a reduction of the relative abundance of others, when those do not grow at equal or higher rate. Therefore one given group could have been increasing, but its share in the sum could appear as if it had instead decreased, if a different group has increased faster. This consideration, applies inevitably for all metagenomics/metabarcoding surveys, and should be kept in mind for all kinds of interpretations about increases and decreases which could be either real or apparent (when driven by a stronger change of a different group). As consequence, comparing different sampling points through time is linked to this inevitable constraint: the compositional nature of the datasets binds all relative frequencies to each other (Gloor et. Al., 2017). Therefore, since, as mentioned, the sum of them is bound to give always 100%, the decrease of a given species could be either apparent (driven by the increase of another), or real (due to its actual negative population dynamics). The problem is that the two causes can not be uncoupled by just comparing species frequencies at the two sampling times. Moreover, as mentioned, even an actual increase of a given species could be masked by the parallel increase of another at a faster pace (or by its net immigration into the scene). For this reason we consider with caution the possibility of looking at taxa fluctuations as indicative of their actual ecological outcomes. Having clarified that we will therefore limit to comment only the major phenomena that stand out from the comparison. The largest taxa trade off that is apparent when comparing the two periods is the decline of Proteobacteria and the parallel rise of Actinobacteria. Trying to frame this within seasonal parameters we can consider that the latter are typically relying on profuse spore formation from colonial growth, while the former are non-sporeforming bacteria either motile via flagella or gliding/swarming mechanisms. The basic life forms of the two groups predict therefore that proteobacteria would be more suited by wet seasons and vice versa. Actinobacteria have been reported by other authors to reach their peaks in fall (Glöckner, et al., 2000). Being also a group of major litter decomposers their rise along with the end of the plants' vegetative season

can be seen as compliant with their landscape and ecosystem cycles. "

Technical corrections:

Line 52: Dispersal not Dispersion. Dispersal is the act itself; dispersion is the result of dispersal.

ANSWER: Corrrection made.

Line 63: Here it seems the topic changes suddenly. Entering deep into the microbiology. I would put the paragraph ending here and not in line 60

ANSWER: Corrrection made.

Line 182: OUT

ANSWER: Corrrection made.

Figure 3: Replace it with a stacked barplot

ANSWER: Fig.3 as anticipated was completely reshaped and increased in font readability, clarity and in-picture legend for taxa colors

Line 310 to 314. This might go up in the methods. Once again, a clear section of the methods describing the methodology is needed.

ANSWER: The methods section has been implemented with the comprehensive new outline described above. One part of this sentence is however to remain in this section as, when recalling the filters operation we describe a result (the intensification of the microbial content in the filtered material during the second 12h lapse,

Line 315: OTUs or taxa? Along all the manuscript I had the feeling these words were not used consistently, please check.

ANSWER: the wording has been corrected where needed. Essentially however, the process of sequencing is yielding reads which are clustered in discrete packages of OTUs, but the subsequent annotation points for each to taxonomical names which are

the ones that we then describe. In this sense, the two terms achieve a formal equivalence. In specific cases (when one taxonomical lineage corresponds to more than one 97% identity-clustered OTUs) we find more correct to use the taxon appellative as it would encompass more than one OTU but they concur to the same best achievable name.

Tab1: It is unclear to me what you are reporting here, please explain better.

ANSWER: This addition to the text that we made will add clarity to the table :

"Besides the comparisons that included all OTUs to put in evidence community variations, in parallel we exploited an additional opportunity to detect possible dust-specific taxa. The rationale was to seek differential enrichment within the dust storm, by dissecting the process, during its progression, splitting its onset from its fully established stage. To this aim we collected separately the filters of the first 12h of the event, and replaced them with new ones that collected air during the second lapse (hours 12 to 24). Thus the availability of two timeframes, both within the dust event, allowed to verify which OTUs would be incrementally enriched along with the progression of the stormy condition. "

Furthermore, we have detailed the legend of the table (now Table 2) to clarify the content. :

" Tab. 2: Extent of OTUs change across cell harvesting time during the same dust outbreak sampling. The percent variation (either increase or decrease) of a given OTUs abundance between the values found in the community obtained by the first 12h sampling and the ones resulting from the following 12h lapse was computed. The average, minimum and maximum percent variation between counts are shown. Only taxa displaying a difference in percentages higher than half of their standard deviation were selected for the present comparison.

Tab2: Are you reporting the number of orders? If it is just the count, that is a measure

of richness, and not diversity. Also, are these results adding important information compared to the bray Curtis similarity table? It seems quite redundant and not helpful for the discussion, since the number of orders is not really helpful for any results. For quantifying community variations, Bray-Curtis is the right instrument, as used later.

ANSWER: Thank you for correcting appropriately the terminology. The term diversity was replaced. The results shown here were considered not equivalent to the Bray-Curtis values, which are relational, as those stem from pairwise operational comparisons involving two samples and expressing distance or similarity; richness is instead a property belonging to a single sample.

Tab3: Simpson and Shannon are both diversity indexes, and the use of two of them does not add much information to the discussion. Why not using richness instead of Simpson?

ANSWER: Although they are estimators of diversity, given the different formulas on which they are based, the use of both could be not redundant. For example in our prior report (Rosselli et al, 2015) on the same island, in the February dust outbreak analyzed at that time, one of the localities (Cagliari) was showing opposite results when using Shannon or Simpson indexes (increasing in one case and decreasing in the other). The richness data are actually shown for all 12 samples in the supplementary dataset at different taxonomical ranks and those regarding the number of orders, as you correctly indicated are also in Tab.2.

Tab4: should be connected to the results at line 259 to 309. But it is not.

ANSWER: The table has been appropriately moved up to the beginning of results and it is now the new Tab.1 and it is discussed along with the taxonomy part of the results

Figure 5A, the PCoA does not add any info to the study. Moreover, the dissimilarity matrix used for the PCoA should be reported.

ANSWER: We agree on this redundancy and PCoA was moved to the supplementary

material as its value was recognized not incremental compared to the PCA, discriminant analysis and the other approaches. The dissimilarity matrix is however available in the manuscript as it is actually in Table 5 (simply, the complement to 1 of those Bray-Curtis similarities, yield the distances upon which the PCoA ordination originates). The form is not that of the canonical triangular matrix as that was rearranged to suit Tab.5 purpose, but all pairwise values are inspectable from the table. Moreover each analysis can be verified and run again from the OTU table which is provided as spreadsheet in the Supplementary dataset S1.

Tab5: This is hard to read. I suggest a heatmap style table.

ANSWER: It is true and we have experimented different ways to reorganize that, including as suggested a heatmap-style arrangement with all the entries ending in a single matrix rather than three separate table sections, which however ended up more complex as it needs a tridimensional representation of the same/different season, dust/control and Cagliari/Sassari location. But what we find more useful to point at, is that the main message that this color table is meant to convey is just the fact that the low similarity pairwise comparison (conditionally formatted as red-yellow shaded cells) are almost all observed in the upper table. i.e. the between-seasons comparisons, while the green ones (higher similarities) are all distributed in the two same-period comparisons. We further stressed this aspect to direct the readers' attention to the way to get the bottom line information from this table. The message is reinforced by the other figures as mentioned but this one shows the Bray-Curtis values and allows each pair to be inspected, which is an information that the PCA, PCoA and dendrogram are not providing with such detail.

565: How was it verified? You would need additional controls after and before the dust events.

ANSWER: We agree, the sentence was actually only meant to imply that if we had observed a very similar situation between the communities of our post-dust sampling

in spring and our pre-dust sampling in fall, one could have hypothesized that the absence of those outbreaks could be conducive to airborne community stability. Such hypothesis would have required other samplings in between to be confirmed as a true stability and not cyclic recurring fluctuations, as you correctly indicate. But as the situation that we observed was in fact the opposite (deep changes occurred without the need of dust outbreaks to cause them), that hypothesis was not the standing one. We therefore limited our comments to notice that the two sites had achieved a high similarity after a dust-free period and that such situation was very different from the one they had displayed four months before. The control for comparison was in this case the last post-dust analysis available, and we stick to this in our inferences.

---

## Author Comment (AC5) · 12 Mar 2021

Dear Reviewers and Editor since the system is not involve nor allow to upload the revised manuscript but just the replies to your queries, I am attaching here a pdf of the new Fig.3.

[Figure]

**Fig. 1.**

[Figure]

---

## Referee Report (RR1)

The manuscript describes a study on the parameters affecting airborne microbial community composition, e.g., season, dust intrusion, geographic proximity to the dust source. These are important questions in the study of aerobiology, especially in the Mediterranean basin that is prone to increasing frequency of Saharan dust intrusions.

My specific remarks are:

L. 20: "…two opposite localities…" – should be "locations".

L. 26: "(118 vs. 65)" – should be moved towards the end of the sentence or removed altogether, at its current location it is unclear what it refers to.

L. 67-68: "Fungal taxa were also analyzed along with their and relationships…" – please correct this sentence.

L. 130-144: This added section describes the samples and their conditions fully and clearly.

L. 143-144: The term "flanking" is somewhat misleading here. The selected sampling events, on a chronological axis are: dust event, clear day, clear day, dust event. There are no samples before the first dust event or after the last one.

L. 147:  There is a stray "." . Please remove.

L. 247: "collection during dust outbreaks under winds from Africa" – the phrasing "under winds" is not correct. Please rephrase.

L. 248: "collection upon under opposite…" – please correct.

L. 252: "add visual aids to each chart attributes" – please remove.

L.257: "their maxima were seen…" – does this refer to all core taxa mentioned, or only to Actinomycetales? Please clarify.

L. 273-274: abrupt line break, please correct.

Table 1: Please add standard deviation to mean values.

Table 2: According to the data presented in this table, the mean variance between the two halves of a single dust event was higher in Cagliari than in Sassari, this is opposed to the number of orders that showed a variance that surpassed the chosen threshold (Table 3). How do the authors interpret each observation?

Table 3: Why show the richness in orders? How is it better than showing the number of OTUs?

L. 377-392: ANOVA is an inadequate statistical test to determine differential abundance. I suggest following the approach presented in Gloor et al., 2017 (Front. Microbiol.), which the authors are familiar with (L. 467), yet for some reason chose not to follow.

L. 424-430: The authors' claim, that an air sample is a built-in average of XXX liters of changing atmosphere, is unreasonable. As the authors themselves state, there is a high day-to-day variability in atmospheric microbiome, yet the hour-by-hour changes

are gradual, as the authors themselves must have noticed when analyzing dust events on a 12h basis. When comparing samples that were obtained by 24h of sampling, it is still important to obtain sufficient replicates of the same scale, for statistical significance. Had one chosen to sample 10 liter of sea water, would it be acceptable to obtain a single sample to represent each condition, claiming that it is an average of 10000 ul of diverse microcosms? Avoiding adequate sampling design by suggesting that the samples themselves represent averages of smaller increments is unacceptable in aerobiology as it is in any environmental microbiome study.

Each environmental sample is somewhat different than its replicate, and some environmental conditions are difficult to replicate, such are dust events. This should encourage deeper and more extensive sampling, and not the opposite. The authors could make the effort to sample more times on clear days, which are not as rare, and their study would only benefit from this choice. In the lack of more samples, it is statistically irresponsible to compare 3 different variables.

L. 467: As the authors cite the problem well defined by Gloor et al. (2017), why not implement the suggested solution to their work? The cited paper gives adequate tools to overcome the compositional nature of sequencing data, yet the authors chose to ignore it altogether.

---

## Referee Report (RR2)

Report BG-2020-324

The authors answered correctly to my concerns and changed their manuscript accordingly.
I accept this paper to be published as it is.

---

## Author Response (AR2)

Dear Dr. Trebs,

we thank you for communicating the updated tasks for the manuscript. One of your comments was related to the presence of colour cells in one of our tables (Tab.5) . We have solved the issue by replacing that table with a figure (Fig. 6). As regards the referees, we acknowledge the fact that Rev #2 has been satisfied by our prior revisions and endorsed the publication of the manuscript. We therefore addressed the remaining issues that concern novel comments from Rev. #3.

We need to add that anonymous Reviewer # 3 has lowered from 'fair' to 'poor' his/her ranking of Scientific Quality Score, in spite of the extensive revision of the manuscript that we had carried out. The revision yielded a 19-page long rebuttal document, which is a rather uncommon case in our experience, and that was moreover in its vast majority actually dedicated to Rev.# 3 issues. This series of additional comments was therefore not expected, also considering that in your own editorial comment of March 16th you had anticipated: *"Your overall responses to the referee comments appear very detailed and complete to me."*

We also noticed that this time the same reviewer has filed the form by checking the option: *"I am not willing to review the revised paper"*. Therefore, our following answers are hereby provided for your attention only. However, we find somewhat uncommon that a person having put forward several novel questions and arguments (and indicating major revisions needed) would disregard hearing whether these have been found appropriate and how authors were responding.

In any event we here below provide the full answers to these novel points.

**Suggestions for revision or reasons for rejection (will be published if the paper is accepted for final publication)**

**The study is of importance to the deciphering of the processes affecting the composition of atmospheric microbiome. It addresses important questions such as the relevance of the time of the year, the occurrence of dust events and the location of the sampling. However, it is my opinion that the small number of samples, compared with the number of examined variables, produces results with low statistical significance, if any.**

ANSWER: in first instance we find surprising the Reviewer commenting that the manuscript *"addresses important questions such as the relevance of the time of the year"*. The reason is because in the first round of reviewing the same reviewer insisted on the fact that our original interpretation of 'seasonal' changes (i.e. meant as relative to the spring vs. fall comparison), were instead to be considered as just 'temporal', with no proven correlation with the time of the year. Therefore, we had accordingly dropped the view of cyclic or recurring events, and accepted that of successional phenomena. The concept had been extensively revised throughout the manuscript, including large parts of the discussion, until the change of the original title, in which, as requested by the reviewer, the reference to the time of the year (season) had been removed. This is "why" we do not understand why the Reviewer is instead praising again the original *"relevance of the time of the year"* having asked to dismiss it.

Second, we found surprising that some basic questions related to the experimental design, as number of samples and number of variables, could still arise at this time of the revision. We had thoroughly addressed the concept of sample and that of control in our prior round of revisions and pointed out that we had analyzed all the dust-outbreak events of that year, so that the 'samples' could not be more than those. And that, given the definitions, the controls could not have been 'any' of the quiet days before or after the storms, but just the last day before and the first day after. Therefore, we simply analyzed all samples available and we did it in two opposite locations.

About the "*low statistical significance if any*" sentence, sounding as a disdainful dismissal of the entire research, we do not understand what the reviewer would be referring to. On the contrary, as regards the whole community level, we showed that the September sampling was significantly different from the May

sampling, in terms of ecological indexes (p Value = 0.00584 for species diversity, and pValue = 0.026 for community evenness); while as regards the single species level, we then showed that 76 taxa were found featuring p values < 0.05. From these, upon applying a stringent Bonferroni-adjusted p-value correction, six of them stood above the significance cutoff, and all within minimal false discovery rate values (FDR < 0.005). All of them were cases highly reduced in September in comparison to May. In the most differentially featured cases, we were able to reach a p-value as low as = 0.000019. Moreover, the robustness of these results was tested by running the analyses independently with parametric as well as non-parametric methods, testing both an ANOVA variance analysis and the non-parametric Wilcoxon Rank test for the verification of the ranking. The two tools gave fully coherent results. The distinctness of our rankings of the variables was moreover confirmed (and shown in the manuscript and supplementary materials), by: Cluster Analysis, Principal Component Analysis; Principal Coordinate Analysis, Linear Discriminant Analysis, Bray Curtis Similarity Analysis of all the 66 pairwise community comparisons;

**It is suggested that the authors focus their effort in comparing a single, at most two, variables (the location and the time) and re-analyze their results accordingly.**

About having considered three variables instead of less (preferably one, as recommended by the reviewer #3). We need to point out three facts; 1) aim (and title) of this paper is "Determining the hierarchical order by which the variables of sampling period, dust outbreaks occurrence, and sampling location, can shape the airborne bacterial communities in the Mediterranean basin". It should be clear that if one wants to rank items into an order, they need to be more than one. 2) What the reviewer is asking about the analysis is indeed what was done. Results were analyzed independently one variable at a time. We compared which variable yielded the highest shifts in pairwise community distances. We did not analyze these data with general linear nor regression models in which one would test interactions between variables. 3) In submitted reports we were used to hear reviewers asking to do more analyses, and we found strange to hear a complaint about having done too many.

**PDF attachment comments:**

**The manuscript describes a study on the parameters affecting airborne microbial community composition, e.g., season, dust intrusion, geographic proximity to the dust source. These are important questions in the study of aerobiology, especially in the Mediterranean basin that is prone to increasing frequency of Saharan dust intrusions.**

**My specific remarks are:**

**L. 20: "…two opposite localities…" – should be "locations".**

ANSWER: Correction done.

**L. 26: "(118 vs. 65)" – should be moved towards the end of the sentence or removed altogether, at its current location it is unclear what it refers to.**

ANSWER: The word orders was moved.

**L. 67-68: "Fungal taxa were also analyzed along with their and relationships…" – please correct this sentence.**

ANSWER: Correction done.

**L. 130-144: This added section describes the samples and their conditions fully and clearly.**

ANSWER: We acknowledge this positive comment.

**L. 143-144: The term "flanking" is somewhat misleading here. The selected sampling events, on a chronological axis are: dust event, clear day, clear day, dust event. There are no samples before the first dust event or after the last one.**

ANSWER: Correction done, the word was removed.

**L. 147: There is a stray "." . Please remove.**

ANSWER: Correction done.

**L. 247: "collection during dust outbreaks under winds from Africa" – the phrasing "under winds" is not correct. Please rephrase.**

ANSWER: Correction done. (…brought by winds from Africa)

**L. 248: "collection upon under opposite…" – please correct.**

ANSWER: Correction done.

**L. 252: "add visual aids to each chart attributes" – please remove.**

ANSWER: Correction done.

**L.257: "their maxima were seen…" – does this refer to all core taxa mentioned, or only to Actinomycetales? Please clarify.**

ANSWER: It referred to all three core taxa. We clarified that in the text.

**L. 273-274: abrupt line break, please correct.**

ANSWER: Correction done.

**Table 1: Please add standard deviation to mean values.**

ANSWER: values have been added.

**Table 2: According to the data presented in this table, the mean variance between the two halves of a single dust event was higher in Cagliari than in Sassari, this is opposed to the number of orders that showed a variance that surpassed the chosen threshold (Table 3). How do the authors interpret each observation?**

ANSWER: As will be clear from the answer to the next query, there is no expectedly proportional correlation between the order and number of OTUs that it could encompass, as some orders could have undergone active diversification and others could be at the opposite and be represented by as little as a single OTU. In this Table we used the OTU unit as we wanted to catch the slightest changes possible in the collected air. In Table 3 we wanted instead to assess the extent of the diversity spectrum in its broadest systematics span.

**Table 3: Why show the richness in orders? How is it better than showing the number of OTUs?**

ANSWER: The OTUs are not reflecting the completeness of bacterial lineages and ecosystem function varieties, as they are simply defined by a clustering cutoff of 97% nucleotide identity. Thus, theoretically, even all different OTUs of a project could even fall within the same single genus. The OTUs richness of a community is more reflective of the micro-evolutionary history of the place than its overall taxonomical latitude. For this reason, a relatively high rank as the order was deemed more apt to convey a true picture of the existing differences among populations.

**L. 377-392: ANOVA is an inadequate statistical test to determine differential abundance. I suggest following the approach presented in Gloor et al., 2017 (Front. Microbiol.), which the authors are familiar with (L. 467), yet for some reason chose not to follow.**

ANSWER: in first instance, ANOVA is a suitable tool to assess differential abundance. This is also testified by the practice of similar countless studies, as well as of being among the standard tools included into the Calypso webtool suite for microbial communities sequencing data comparisons (Zakrzewski, et al., Bioinformatics 33, 782–783,2017), which has, to date, used and cited by 292 articles indexed in Web of Science.
Moreover, since the only concern with the use of ANOVA regards its well-known basic requirements of distribution normality and variance equality, in our manuscript we had also provided the results of a parallel analysis of the very same data, independently done with the non-parametric alternative to ANOVA, the Wilcoxon Rank Test. That means, that we verified each result with a procedure that does not involve parameters as means and standard deviation and, as stated in each of the versions of the manuscript, we obtained matching results in terms of significantly different taxa. This was already clearly written in the same lines that the reviewer has signaled. Quoting our manuscript: "In order to determine which bacterial taxa were mostly accompanying/causing those changes in a statistically significant manner, and to rank their individual importance in this phenomenon, we run an analysis of the differentially featured taxa, testing both an ANOVA variance analysis and a non-parametric Wilcoxon Rank test verification of the ranking. The two tools gave coherent scores and the results of the ANOVA output are shown in Supplementary Table S1."
As regards the mention to the approach described by Gloor et al. 2017, the reviewer has assumed that we have not taken that into consideration, but instead we had. In addition to the square root transformation, we had verified also the Aitchison's centered log-ratio transformation (CLR), and compared the results of each transformation method. Notwithstanding some slight changes in the shape of the ordination plots, observed phenomena and ensuing trends that we point out and describe in this report were found to be exactly the same. The caveats signaled by Gloor et al., 2017 imply that some communities may suffer from compositional constraints, and some communities could be minimally affected. It depends on the community structure, and does not imply that every author should only construct graphs based on centered log ratio from now on. The reasons for which, once verified the output equivalences with CLR, we opted for showing the former transformation procedure (TSS) is that the combination of total sum scaling with square root transformation is the renowned 'Hellinger transformation'. This transformation has been praised as a preferable choice in ecological community comparisons, (Legendre and Legendre, 1998, Numerical ecology, 2nd English edn. Elsevier, Amsterdam), as it offers the best trade-off between linearity and resolution in comparison to chi-square metrics and other approaches. It is also recognized as more balanced for the weight given to rare species. For this reason it is also the first recommended choice of data transformation in the Calypso webtool suite (Zakrzewski, et al., Bioinformatics 33, 782–783,2017), which, as mentioned above, has to date used and cited by 292 articles indexed in Web of Science. Therefore, once we ruled out the risk that the compositional nature of the datasets could affect the results (by testing CLR as well), we opted for the Hellinger method. We need to remark that this explanation had been already given to a different Reviewer (Rev. #2) in our prior 'Answers to reviewers' document.

**L. 424-430: The authors' claim, that an air sample is a built-in average of XXX liters of changing atmosphere, is unreasonable. As the authors themselves state, there is a high day-to-day variability in atmospheric microbiome, yet the hour-by-hour changes are gradual, as the authors themselves must have noticed when analyzing dust events on a 12h basis. When comparing samples that were obtained by 24h of sampling, it is still important to obtain sufficient replicates of the same scale, for statistical significance. Had one chosen to sample 10 liter of sea water, would it be acceptable to obtain a single sample to represent each condition, claiming that it is an average of 10000 ul of diverse microcosms?**

ANSWER: No, it is the opposite. And exactly by the same argument that the reviewer points out: since there is a continuous hour-by-hour change, sampling for short times instead of sampling for 24 hours, would not yield replicates but different samples representing successional frames of the evolving new condition, which would not be legitimate to treat as replicates and to compare. Only if one could afford the expense of 24 Skypost Tecora motorized devices, it could be possible to filter simultaneously and in parallel, air for one hour in the same location and obtain truly legitimate replicates. Besides, the variable that we considered here is the dust outbreak, which, as such, exists in two possible statuses: ON or OFF. Therefore the 24 h sampling is carried out completely in the presence of the dust event and indeed not only accounts for the possible variation that takes place in it, but definitely seeks to incorporate as much as possible of that to be more adequately representative of that condition. The fact that variation occurs is also demonstrated by our first-12h/second-12h variation. The concept we are relying on, is exactly the same that is used in soil analyses. In these cases, a plot is analyzed by tracing a cross and pooling five subsamples collected at the center and at each of the four corners transect to obtain a single sample that incorporate as much as possible the spatial variability. And even the example of 10 liters of water chosen by the reviewer is not different, since if one is not sequencing 10 liters but small aliquots, the possibilities are 1) using some of these aliquots to analyze them and consider them as replicates; or 2) Filter all of those 10 liters and sequence the resulting concentrated community. In the first case if one takes for example 10 replicates of 1 ml each one would have analyzed 1 /10000$^{th}$ of the sampled environment. In the second case, filtering all as we did with the air, and resuspending the filtered cells in 1 ml, one would analyze the whole community. Contrary to the reviewer affirmation, our single sample is not "an average of 10000 ul of diverse microcosms". In fact it is not an average at all, as it is the totality. (By the way, in 10 liters there are not 10000 ul but 10000000 ul.)

**Avoiding adequate sampling design by suggesting that the samples themselves represent averages of smaller increments is unacceptable in aerobiology as it is in any environmental microbiome study.**

ANSWER: No, besides the fact that in aerobiology, sampling through a continuous time lapse as we did is the standard practice, as just answered above. We did not do any averaging here as we analyzed the whole sampled amount.

**Each environmental sample is somewhat different than its replicate, and some environmental conditions are difficult to replicate, such are dust events. This should encourage deeper and more extensive sampling, and not the opposite.**

ANSWER: There is an open and naïve contradiction in this sentence: "Each environmental sample is somewhat different than its replicate". If a sample is acknowledged in the first place, as 'already known to be different from another sample', that can never be taken as 'replicate'. Replicates are legitimately assumed as samples which obey to the assumption of being comparable as belonging to the same experimental condition. If one already knows that an incremental gradient of variability among replicates is predetermined, the legitimate choice is to pool them into a single sample that represents the persisting overall condition, in our case the 'ON' state of the dust discharge.

**The authors could make the effort to sample more times on clear days, which are not as rare, and their study would only benefit from this choice.**

ANSWER: We had actually already addressed this in the prior round of revisions and we state that again here:
Since it is already known from literature that, as microbial community composition is concerned, even in the absence of dust outbreaks, the ambient state of the atmosphere is not stable either, our goal was not to compare an hypothetical *status quo* with an altered one. The meaning of the 'control' here was to catch 1) the first possible timeframe after the stopping of a northbound dusty wind outbreak (it occurred in May) or 2) the latest possible timeframe of a situation before the onset of a dust-carrying change of wind regime (it occurred in September). Thence, in the latter event the control is not intended as a situation of calm that could represent a period of unknown length, but rather the time-zero sample of the dust event itself. While for the former case in May, the control is symmetrically designed as the quiet after the storm.

**In the lack of more samples, it is statistically irresponsible to compare 3 different variables.**

ANSWER: Asa already commented above. The reviewer is apparently missing the point that we are comparing the variables independently one at a time and we are not testing their interaction in a linear model or any other way.

**L. 467: As the authors cite the problem well defined by Gloor et al. (2017), why not implement the suggested solution to their work? The cited paper gives adequate tools to overcome the compositional nature of sequencing data, yet the authors chose to ignore it altogether**

ANSWER: We just paste-repeat the answer given above:
As regards the mention to the approach described by Gloor et al. 2017, the reviewer has assumed that we have not taken that into consideration, but instead we had. In addition to the square root transformation, we had verified also the Aitchison's centered log-ratio transformation (CLR), and compared the results of each transformation method. Notwithstanding some slight changes in the shape of the ordination plots, observed phenomena and ensuing trends that we point out and describe in this report were found to be exactly the same. The caveats signaled by Gloor et al., 2017 imply that some communities may suffer from compositional constraints, and some communities could be minimally affected. It depends on the community structure, and does not imply that every author should only construct graphs based on centered log ratio from now on. The reasons for which, once verified the output equivalences with CLR, we opted for showing the former transformation procedure (TSS) is that the combination of total sum scaling with square root transformation is the renowned 'Hellinger transformation'. This transformation has been praised as a preferable choice in ecological community comparisons, (Legendre and Legendre, 1998, Numerical ecology, 2nd English edn. Elsevier, Amsterdam), as it offers the best trade-off between linearity and resolution in comparison to chi-square metrics and other approaches. It is also recognized as more balanced for the weight given to rare species. For this reason it is also the first recommended choice of data transformation in the Calypso webtool suite (Zakrzewski, et al., Bioinformatics 33, 782–783,2017), which, as mentioned above, has to date used and cited by 292 articles indexed in Web of Science. Therefore, once we ruled out the risk that the compositional nature of the datasets could affect the results (by testing CLR as well), we opted for the Hellinger method. We need to remark that this explanation had been already given to a different Reviewer (Rev. #2) in our prior 'Answers to reviewers' document.

Counting on having clarified all the pending issues we thank you for your kind attention and cooperation.

Andrea Squartini

---

## Author Response (AR3)

Associate Editor Decision: Publish subject to technical corrections (10 Jun 2021) by Ivonne Trebs
Comments to the Author:

Dear authors,

thank you very much for the responses and rebuttal of the referee comments. Before publication of your final manuscript, I recommend to add some more details (2-3 sentences) of your explanations given in the response letter about the statistical approaches (in particular Aitchison's centered log-ratio transformation). Please also check that the letter size in your Figure files is large enough (also in the Supplement).

Best regards,

Ivonne Trebs

Dear Dr. Trebs,
Thank you for your remark, we have introduced the following sentences:

In Materials and Methods:

"The resulting ordination results were also compared with those yielded by the alternative Centered Log Ratio (CLR) data transformation. "

In Discussion:

"In order to explore the extent of this possible bias we compared the results of the ordination plots yielded by two data transformation procedures; in addition to the TSS square root transformation, we checked also the Aitchison's centered log-ratio transformation (CLR). Notwithstanding some slight changes in the shape of the ordination plots, observed phenomena and ensuing trends that we point out and describe in this report were found to be the same. The reasons for which, we opted for showing the former transformation procedure (TSS) is that the combination of total sum scaling with square root transformation is the renowned 'Hellinger transformation'. This has been praised as a preferable choice in ecological community comparisons, (Legendre and Legendre 1998), as it offers the best trade-off between linearity and resolution in comparison to chi-square metrics and other approaches. It is also recognized as more balanced for the weight given to rare species. "

The reference from Legendre and Legendre has been added to the references.

The letter size has been inspected and deemed satisfactory

Looking forward to your decision
Best wishes

Andrea Squartini